biomathematics

ascertainment rate, testing, COVID-19, outbreak size

**Authors for correspondence:**
C. Browne
e-mail: cameron.browne@louisiana.edu
H. Gulbudak
e-mail: hayriye.gulbudak@louisiana.edu

# Modelling COVID-19 outbreaks in USA with distinct testing, lockdown speed and fatigue rates

## J. C. Macdonald, C. Browne and H. Gulbudak

Department of Mathematics, University of Louisiana at Lafayette, Lafayette,
LA 70503-2014 USA

JCM, 0000-0002-3643-6266; HG, 0000-0002-4397-1494

Each state in the USA exhibited a unique response to the COVID-19 outbreak, along with variable levels of testing, leading to different actual case burdens in the country. In this study, via *per capita* testing dependent ascertainment rates, along with case and death data, we fit a minimal epidemic model for each state. We estimate infection-level responsive lockdown/self-quarantine entry and exit rates (representing government and behavioural reaction), along with the true number of cases as of 31 May 2020. Ultimately, we provide error-corrected estimates for commonly used metrics such as infection fatality ratio and overall case ascertainment for all 55 states and territories considered, along with the USA in aggregate, in order to correlate outbreak severity with first wave intervention attributes and suggest potential management strategies for future outbreaks. We observe a theoretically predicted inverse proportionality relation between outbreak size and lockdown rate, with scale dependent on the underlying reproduction number and simulations suggesting a critical population quarantine 'half-life' of 30 days independent of other model parameters.

## 1. Introduction

The COVID-19 pandemic began in Wuhan, China and spread rapidly throughout the world in early 2020 provoking a wide range of interventions. China gained rapid control of its epidemic, probably due to its rapid and strict lockdown as discussed in several studies [1–3], in contrast to several other countries with slower and less unified efforts. In particular, the USA employed a heterogeneous response, with different scales of quarantine measures (stay-at-home orders and business closures), despite the first reported case of COVID-19 occurring on 21 January 2020 in Washington state and cases being reported in all 50 states by mid-March [4]. Furthermore, each state displayed

distinct exit strategies and lockdown fatigue, and corresponding fatigue half-life (time until 50% of population which has entered self-quarantine/lockdown returns to normalcy) even though cases never were brought down to low levels on a national scale. Another feature of the US response was variable testing through time and by state, growing from sparsely to more widely available tests, leading to distinct increasing trends of case detection.

Each state represents distinct realizations of how different response characteristics and case ascertainment correlated to true case burden in the United States, which provides fertile ground for testing outbreak containment strategies. However, successfully fitting an epidemic model to this heterogeneous response exhibited in the first wave COVID-19 outbreak in the USA presents a number of challenges. The multitude of epidemic curve shapes induced by the distinct dynamic behavioural and government interventions can lead to issues of over-fitting or a large number of parameters that blur the most important factors [5]. In addition, the unknown true number of cases calls for methods to incorporate testing, mortality or other sources of data (e.g. seroprevalence studies), which is complicated by changing quantities, such as detection [6–8], or antibody levels [8]. Here, we focus on a unified model both simple and flexible enough to fit the wide range of COVID-19 outcomes in the USA (to 31 May 2020), which incorporates the critical factors in epidemic trajectory. Furthermore, we construct an appropriate *per capita* testing dependent ascertainment rate calibrated with mortality data to both allow more accurate model fits and provide a means of estimating the actual number of cases.

The wide range of (self) quarantine responses in the different states and resultant outcomes allow us to analyse potential responses to future outbreaks using correlation and sensitivity analyses as well as counterfactual simulation studies. In this way, our minimal number of identifiable parameter values can be compared among the states representing crucial control quantities, such as lockdown speed and fatigue half-life, and via analytically derived relations, we link outbreak size as inversely proportional to population (self) quarantine rate. We also provide estimates of commonly used quantifiers of outbreak severity, epidemic trajectory and effectiveness of control measures, such as infection fatality ratio (IFR) and the ratio of (estimated) true cases to reported cases, accounting for common sources of error in their prediction [9,10]. By estimating these quantities for all states and territories under one modelling framework we provide a comprehensive overview of the first wave outbreak in the USA. Ultimately we synthesize these results into a range of potential future responses which highlight the critical nature of widespread, swift quarantine measures, with a model suggested critical duration, for a rapidly spreading outbreak. These results can be used to inform management strategies both in locales where COVID-19 vaccines are not yet widely available, and for future outbreaks of emergent pathogens.

## 2. Model

First, we formulate an epidemic model which can account for the heterogeneous response to COVID-19 outbreaks exhibited in the USA. In [1], we developed an SIR-type model applied to the outbreak in China incorporating terms for responsive self-quarantine (lockdowns) and contact tracing, where the rate of both control actions depend upon current infection rates. Here, we focus our modelling framework to fit the wide range of reactionary lockdown measures and testing observed in different states during the first wave of COVID-19. Consider the following system (see also figure 1) for susceptible ($S$) and quarantined (or socially distanced) susceptible ($S_q$) population, and infected ($I$) individuals who ultimately progress to reported ($R$), unreported ($U$), or dead ($D$) cases.

$$
\left.
\begin{aligned}
\frac{\mathrm{d}S(t)}{\mathrm{d}t} &= -(1 + \psi)\beta S(t)I(t)/N + \alpha S_q(t), \\
\frac{\mathrm{d}S_q(t)}{\mathrm{d}t} &= \psi\beta S(t)I(t)/N - \alpha S_q(t), \\
\frac{\mathrm{d}I(t)}{\mathrm{d}t} &= \beta S(t)I(t)/N - \frac{1}{T}I(t), \\
\frac{\mathrm{d}R(t)}{\mathrm{d}t} &= \frac{\rho(t)}{T}I(t), \\
\frac{\mathrm{d}U(t)}{\mathrm{d}t} &= \frac{1 - \rho(t)}{T}I(t) \\
D(t) &= \xi \int_0^t \frac{a^{\eta-1}\exp(-\eta a/\mu)}{\Gamma(\eta)(\mu/\eta)^\eta} \cdot \frac{\beta}{N}I(t - a)S(t - a)\,\mathrm{d}a.
\end{aligned}
\right\}
\tag{2.1}
$$

and

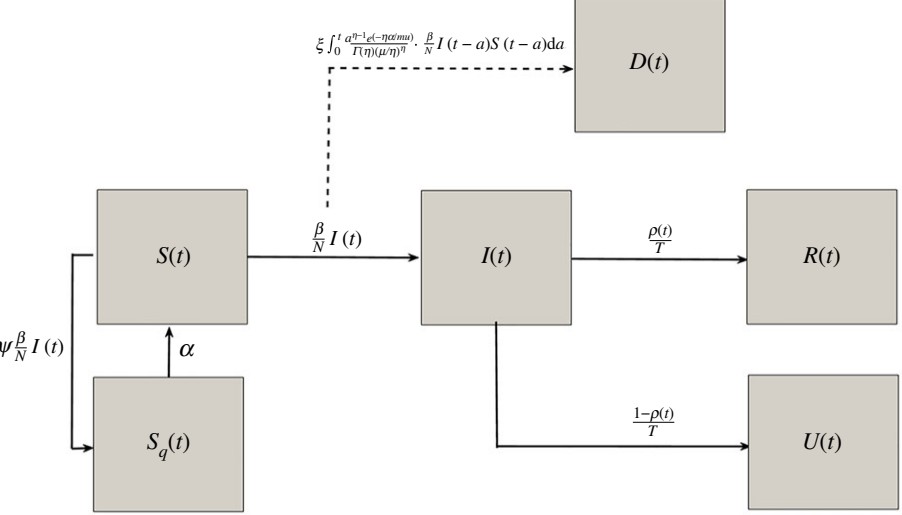

**Figure 1.** Model compartments are susceptible individuals ($S$), self-quarantined individuals ($S_q$), infected individuals ($I$), reported cases ($R$), unreported cases ($U$) and deaths ($D$), see also table 1. Note that the solid lines and corresponding model terms correspond to *per capita* transition *rates*, while the cumulative death *amount* term is indicated by the dashed line.

Our model assumes susceptible individuals become infected relative to force of infection $\lambda(t) = \beta I(t)/N$, where $\beta$ is the transmission rate and $N$ is the population size. We model the primary mechanism of lockdown/self-quarantine—sheltering large portions of susceptible population—by having susceptible individuals transition to quarantined state $S_q$ at rate $\psi \lambda(t)$, where $\psi$ is a proportionality constant with respect to force of infection $\lambda(t)$, denoted as the *self-quarantine (rate) factor*. The product of $\psi$ with force of infection is a phenomenological relation between lockdown/self-quarantine rates and current infection levels. Indeed, several works [11–15] have shown that population activity measures (e.g. mobility, economic transactions, percentage of people staying/working at home) were primarily driven by individual reaction to media and perceived risk tied to COVID-19 case incidence, and secondarily influenced by government mandates. State lockdown orders are also inherently related to case counts, but may be enacted in a temporally discrete manner with other factors affecting their proclamation. Although reporting accuracy and delays in response complicate the relationship of human behavioural changes and government action with raw infection incidence, our formulation offers a simple measure of population self-quarantine rate relative to case incidence (see [1] for re-scaled rates accounting for reporting delay).

While individuals are in self-quarantine it is assumed that they do not contact with infected population, and that individuals exit quarantine with rate $\alpha$ *lockdown fatigue*. On the population level, $\alpha$ measures *lockdown fatigue*, in other words the tendency for individuals and government to revert to normal regardless of infection level after a certain amount of time. Even though $\alpha$ is not linked to force of infection, the incidence levels will inevitably drop after lockdown (dependent on magnitude of $\psi$), so population will revert to normalcy when the disease is more under control than before lockdown/self-quarantine (given the range of $\psi$ and $\alpha$ parameters fitted for each state detailed in next section). Other approaches for capturing large-scale social distancing/self-quarantine in populations have been used, such as assuming time-dependent transmission/contact rates ($\beta(t)$) [16,17] or considering constant rate of susceptible transition to quarantined state [18]. While there are advantages/disadvantages to each modelling approach, we contend that our nonlinear social distancing rate captures a contagion-like behavioural response to rising infected cases, and allows us to derive novel formulae for final and peak outbreak size (see §4). Furthermore, by tying quarantine to new infection rate, we capture the observed rapid large-scale response of varying strength across states, which saturates and wanes as cases drop and fatigue sets in, mimicking mobility data from [11] (figure 7). Our model also captures the social nature of $\psi$ in the positive relationship between $\psi$ and $\mathcal{R}_0$, discussed further in the Results section (figure 5).

Infected individuals either have their cases reported (model compartment $R$), or fail to have their cases reported (model compartment $U$), with mean time until case is reported $T = 7.5$ days, with the proportion of individuals in each compartment determined by ascertainment rate $\rho(t)$, or the proportion of true cases at a given time captured by testing. Regardless of report status death is expected to occur with infection fatality ratio (IFR), represented by model parameter $\xi$, following a gamma-distributed delay after infection with mean of $\mu = 21$ days, based upon [19]. The model parameters and descriptions are also listed in table 1. In addition to the fixed parameter values for

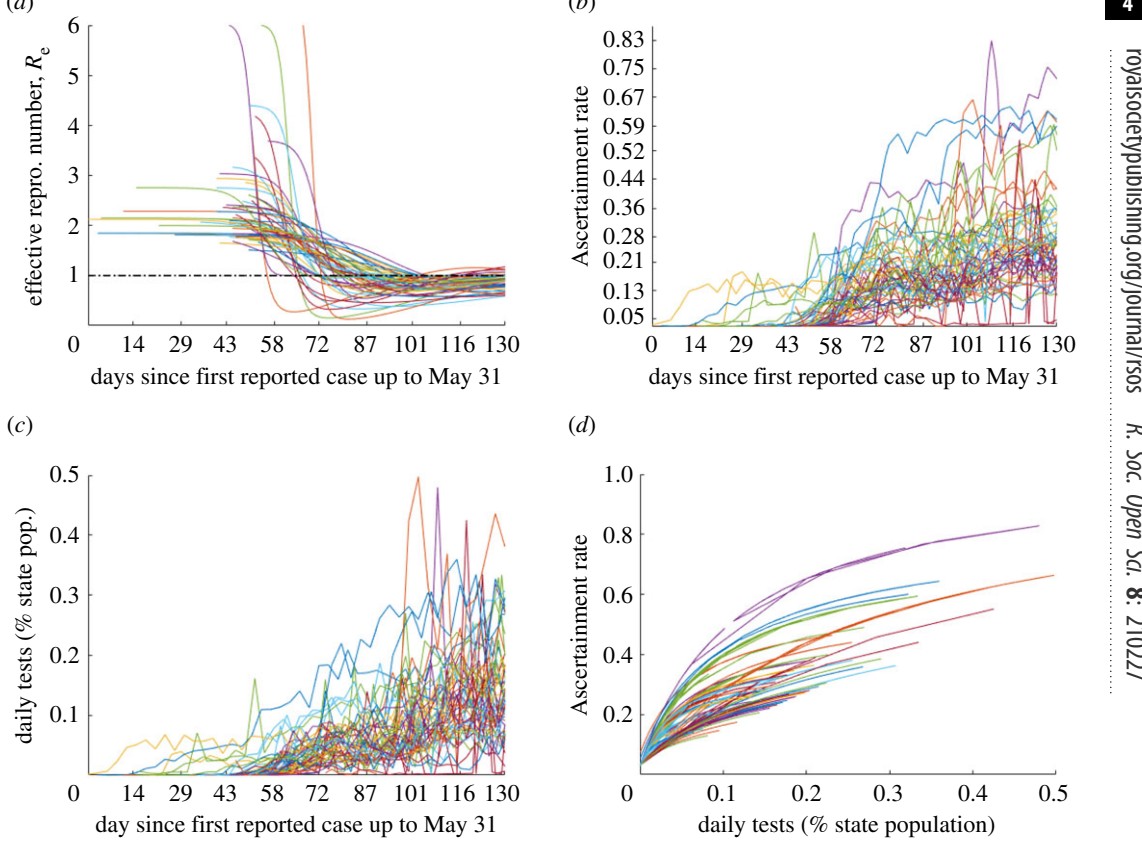

**Figure 2.** (*a*) Time variable reproduction number; $\mathcal{R}_e(t)$, states with later reported first case have higher basic reproduction number $\mathcal{R}_0 = \mathcal{R}_e(0)$. (*b*) Time variable ascertainment rate. (*c*) Tests as per cent of individual states population. (*d*) Ascertainment rate as variable of daily tests, ascertainment saturates as daily tests increase.

mean time until case is reported, *T*, and mean time until death, *μ*, we fit the remaining parameters of the model as described in the next section, obtaining good fits for the data from all 50 states, Washington DC Guam, Puerto Rico, the Northern Mariana Islands and the US Virgin Islands (see figure 8; electronic supplementary material, figures S1–S14).

## 3. Methods

We now describe the methods of our fitting procedure and analysis of the model. All relevant data and code are available at [22]. Where shown, indicated stay-at-home orders are based upon [23]. Population levels are from US Census Bureau 2019 projections [20]. To obtain parameter estimates for our model in each state and territory, we used cumulative death and case totals [24] as well as daily testing data, inferred from [21]. The data used for the whole US fit was obtained by combining data from all subsidiary states and territories due to concerns about data reliability at the federal level [25]. Even at the state level, the raw testing data contained a number of reporting irregularities, as evidenced by several days with negative daily testing instance. To account for this, we smoothed the data by averaging with nearest neighbours in such a way as to not change the cumulative testing totals prior to finding the moving average.

We define the ascertainment rate function $\rho(t)$ as

$$\rho(t) = \frac{k(\tau(t)/N)}{A/N + \tau(t)/N} + k_0, \tag{3.1}$$

where $\tau(t)$ is the 3-day moving average of the raw daily testing totals, with parameters *k* as maximum ascertainment rate, *A* as half saturation point in terms of tests, and $k_0$ as minimum ascertainment rate. Observe that $\rho(t)$ is chosen to be a saturating function of daily tests as a proportion of population (see figures 2b–d and 3e), and that $\rho$ has an inversely proportional relationship with test positivity (see the

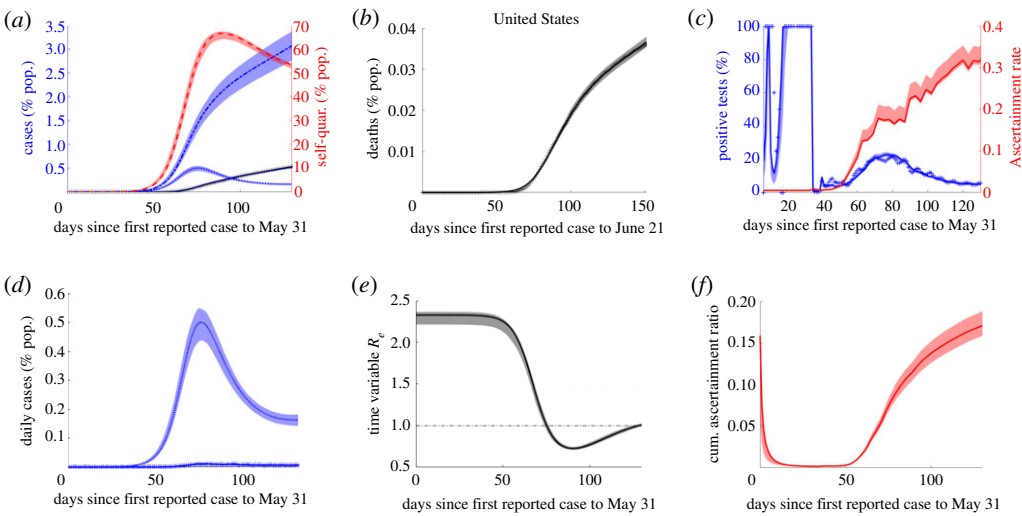

**Figure 3.** Model baseline fit (lines) together with 95% confidence intervals (shaded regions) for USA.

rightmost column of figure 8 and figure 3c; electronic supplementary material, figures S1–S14). As the number of tests increased, so did the ascertainment rate, because the increase in tests was not driven by more demand solely based on rising case numbers, but wider availability and convenience leading more individuals overall to take the tests. The negative concavity of $\rho$ as a function of number of tests reflects the principle of diminishing returns associated with wait times, and also a larger influence of demand due to actual rising cases once tests become widely available, which together lead to saturation of $\rho$ to a maximum level. While this form of $\rho(t)$ is a phenomenological relationship with $\tau(t)$ and there is no formal derivation, we can sketch further justification of a saturating $\rho(t)$ when the number of tests, $\tau(t)$, increases with $t$, as observed over the time frame considered. Indeed, we propose the informal differential equation for $\rho(t)$,

$$\frac{d\rho}{dt} = a\tau(t) - b\rho(t) - \frac{\rho(t)}{T}\frac{d}{dt}\left(\frac{I(t)}{\tau(t)}\right),$$

reflecting increase due to testing availability (at first increasing with testing but linear decreasing as higher proportion tested results in complacency and/or longer waits) and the change in $\rho(t)$ from the flux in positivity rate, respectively. Under appropriate conditions, then $\rho(t)$ will increase to a limit in saturating form as $t \to \infty$.

Additionally, we note that resulting cumulative ascertainment ratio,

$$A_t = \frac{\int_0^t (\rho(t)/T)I(t)dt}{\int_0^t \lambda(t)S(t)\,dt}, \tag{3.2}$$

for Connecticut matches very well with a similar figure in [8] (see electronic supplementary material, figure S15). It is assumed even absent or at low testing, there is a minimal ascertainment rate calibrated based upon low ascertainment levels during the early stages of an outbreak with bounds of 0.005 and 0.03 for the whole of the USA and its territories, and for each state or territory, respectively. This difference in lower bound was chosen because, as indicated by figure 5a, many states' outbreak began significantly before their first reported case, thus when considering the USA as a whole, few cases were ascertained in the initial phase of the outbreak. For each state and territory fit and the whole US fit, the maximum ascertainment rate was fixed based upon the maximum daily test count as a per cent of the states population (table 1). This bound was set based in part on the mean cumulative ascertainment ratio for Connecticut and New York City at the end of May 2020 provided in [8]. For the USA, the bound was set based upon combined model estimated cumulative case total for all states and territories. The half saturation constants were fit to each state, calibrated by the simultaneous fit of mortality and reported cases, to reflect state-specific relationships between testing and ascertainment (figure 2). The obtained ratios of total to reported cases (table 2) are in line with CDC estimates [6,23].

**Table 1.** Key model parameters and quantities with description.

| quantity | description | notes |
|---|---|---|
| $\lambda(t) = \beta I(t)/N$ | force of infection | $N$ from [20] |
| $\psi$ | lockdown (rate) factor | range: 0–1000 |
| $\alpha$ | self-quarantine exit rate (lockdown fatigue) | range: $1/200$–$1/7$ days$^{-1}$ |
| $\log(2)/\alpha$ | time for 50% of pop. who enter self-quar. to return to normalcy (fatigue half-life) | range  'low:' $[5,15)$ days 'intermediate:' $[15, 60)$ days 'high:' $[60, 139]$ days |
| $T$ | mean time until case reported | fixed at $T = 7.5$ days |
| $\rho(t) = ((k(\tau(t)/N))/$  $(A/N + \tau(t)/N)) + k_0$ | daily variable ascertainment rate | testing data, $\tau(t)$, from [21] |
| $k$ | maximum ascertainment rate | fixed for state $i$ at: $(\max{}_t \tau_i(t))/(0.00454N)$ for USA: 0.587 |
| $A$ | half saturation point for testing | range for state $i$: $0.3 \max_t \tau_i(t)$–$0.8 \max_t \tau_i(t)$ tests |
| $k_0$ | minimum ascertainment rate | $k_0 = 0.03$ for states, $k_0 = 0.005$ for USA |
| $\mu$ | mean time until death | fixed at $\mu = 21$ days based upon [19] |
| $\eta$ | gamma shape parameter | range: 0.5–25 |
| $\xi$ | infection fatality ratio (IFR) | feasibility range: 0.001–0.05 |

We numerically solved the nonlinear least-squares problem, using the interior-reflexive Newton method as implemented by MatLab's lsqcurvefit function, minimizing objective function

$$J(t) = \omega_1 R(t) + \omega_2 D(t), \tag{3.3}$$

where

$$\omega_1 = [R(t_f^R)]^{-1} \quad \text{and} \quad \omega_2 = \ell[D(t_f^D)]^{-1} \tag{3.4}$$

were the chosen positive weights for each state, and $t_f^R$, $t_f^D$ are the reported cumulative case and death totals on 31 May and 21 June 2020, respectively. For New York $\ell = 1.25$, for Iowa $\ell = 0.75$, for Mississippi $\ell = 1.0$, for North Dakota $\ell = 5.91$ (after running a search algorithm for $\ell \in [0.75, 8]$ while retaining visual quality of fits, for where average relative error (ARE) for $\alpha$ was minimal), for all other states and territories $\ell = 2.5$, and for the USA $\ell = 2.75$. For each state or territory, the time interval considered was from the first confirmed case up until 31 May 2020 for cases, by which time most states' mandated stay-at-home orders had ended [6,23], and 21 June 2020 for deaths. For our fittings, the mean time until case is reported was fixed at $T = 7.5$ days and the mean to death at $\mu = 21$ days to improve model identifiability. These values are in line with the weighted (by proportion of total cases) mean parameter values obtained from fitting the model without fixing these parameters (see program files on GitHub). All other model parameters, as well as $I_0$, the initial case total, were fit for each state or territory. The fit values (see table 2 and the electronic supplementary material for all parameter values) were then used to provide approximate true cumulative case totals for the time period considered

$$\mathcal{C}_t = \int_0^t \lambda(t)S(t)\, dt, \tag{3.5}$$

where $\lambda(t)$ is the force of infection (table 1). We found that fitting both case ascertainment, $\rho$, and infection fatality ratio, $\xi$ (bounded within a feasible range from 0.001 to 0.05), were necessary to obtain good fits across all states and territories.

In order to compare *% model predicted positives* (MPP) with *% positives inferred from the data* (PID) and to highlight the relationship between test positivity and $\rho(t)$, for each state and territory, we plotted these three quantities together,

$$\text{PID} = 100\left(\frac{d(t)}{\tau(t)}\right) \quad \text{and} \quad \text{MPP} = 100\left(\frac{\delta(t)}{\tau(t)}\right), \tag{3.6}$$

where $d(t)$ is the 3-day rolling average of case totals inferred from reported cumulative case totals after applying the same smoothing to daily case totals as was applied to daily testing incidence, and $\delta(t)$ is the daily reported case incidence inferred from the fit cumulative case totals (figures 3 and 8).

For all states and territories, with the exception of North Dakota when comparing $\alpha$ (due to this being the only parameter not practically identifiable across all states and territories), the relationship between model parameters and outbreak trajectory was assessed. The reproduction number $\mathcal{R}_0$ can be used as a single parameter. Indeed, for model (2.1),

$$\mathcal{R}_0 = \beta T, \tag{3.7}$$

(noting that the susceptible quarantine does not affect $\mathcal{R}_0$ because it is proportional to force of infection). Then since $T$ is fixed (at 7.5 days), the quantity $\mathcal{R}_0$ can be identified with parameter $\beta$. We computed time and test variable ascertainment rate, $\rho(\tau(t))$ as well as time variable reproduction number, $\mathcal{R}_e = (S/N)\mathcal{R}_0$ (figure 2). We also analysed the association of estimated parameter values with outbreak measures, along with correlation between parameters, using Spearman's rank correlation (see figure 5; electronic supplementary material, figure S19 for statistically significant results).

To assess identifiability and confidence in our fitting procedure, we conducted uncertainty analysis. Similarly to [26], this analysis was carried out in the following manner (note we assume normal error structure instead of Poisson error structure because the Poisson assumption generates datasets with less variation than the original data (see electronic supplementary material, figure S17)):

  (i) Simultaneously fit the model (2.1) to cumulative case and death totals, and testing data as described above.
 (ii) Obtain the inferred fit daily case and daily death curves, and under the assumption that the reporting error is normally distributed and relative in magnitude to the reported total at each data point,

$$y_i = g(x(t_i), \hat{\theta}) + \epsilon_i \quad \epsilon_i \sim n(0, y_i \cdot s^2),$$

where $g$ is the true number of daily cases (deaths) and $\hat{\theta}$ the set of true parameter values.
(iii) Generate 200 daily case (death) datasets with noise levels of 40% and 70% for the USA and individual states, respectively; we then summed these daily case results to generate cumulative case (death) datasets and simultaneously refit the model to each of these generated datasets. These noise levels were used because for the USA this value causes the generated datasets to cover the original data (see electronic supplementary material, figure S17), and for individual states a higher noise level was used to account for greater variability in the segregated data.
(iv) Arrange the generated values in increasing order and remove the top and bottom 2.5% in order to obtain the desired approximate 95% confidence intervals.

The resulting fit, confidence intervals, and average relative errors (see figure 3 and table 5 for the USA, and tables 3 and 4 for individual states) indicate that the key model parameters are all practically identifiable except for $\alpha$ in one state, North Dakota, assuming assigned noise levels in daily case (death) reporting. Thus North Dakota was excluded from any correlation analysis involving $\alpha$ (or half-life fatigue).

In addition, to further examine the relationship between $\psi$, $\mathcal{R}_0$, $\alpha$ and outbreak trajectories, we also highlight several states with different relative relationships between $\psi$ and $\alpha$ representing the range of outbreak responses in the USA, and conduct both sensitivity analysis and counterfactual simulation studies with these examples (see figures 9–11; electronic supplementary material, figure S16). We fixed model parameters as those obtained from our fitting process, and varied $\mathcal{R}_0$, $\psi$ and $\alpha$ to examine their differential impacts on epidemic trajectory. The fit plots for all states and territories are included in the electronic supplementary material.

# 4. Results

Our unified model simultaneously fits each state's cumulative mortality and reported case data, along with daily per cent positive tests through our calibrated ascertainment rate, starting at first reported case to 31 May 2020 (as described in Methods). The estimated parameters, particularly quarantine (rate) factor, $\psi$, fatigue half-life, $\log(2)\alpha^{-1}$, reproduction number, $\mathcal{R}_0$, and infection fatality ratio, $\xi$, provide key information about the variability of COVID-19 spread and response in the USA. The resulting parameter values (table 2) and associated fits provide true cumulative case estimates for all 55 states and territories considered. These range in value from as low as 0.30% (confidence interval 0.25–0.39) of population (Hawaii) to as high as 10.06% (CI (8.57–12.78) of population (New York State) over the time period, with an estimated 3.07% of the entire US population having been infected as of 31 May 2020 (CI 2.76–3.37% population). These values represent both the wide range of observed outcomes and highlight the importance of incorporating testing data to obtain sufficient model flexibility (figures 3 and 8; electronic supplementary material, figures S1–S14). Indeed, the ratio of true cumulative cases to reported cases across states ranged from as low as 2.24 (Rhode Island, CI 2.05–2.37) to as high as 17.70 (Puerto Rico, CI 16.12–19.23) with a value of 5.85 (CI 3.51–6.30) for the USA, in line with CDC estimates [23]. Our fit IFR, both for every state and for the entire USA, works to counter both a common source of overestimation and a common source of underestimation [9,10]. Indeed, our gamma distributed delay (mean $\mu = 21$) of days after infection until death and fitted true cumulative case totals with ascertainment rate $\rho(t)$, together estimate IFR ranging from as low as 0.001 (Guam, CI 0.0010–0.0014) to as high as 0.0283 (New Jersey, CI 0.024-0.035). There were outbreaks from mild to severe toll; with the fit IFR for the USA being 0.012 (CI 0.0108–0.0133) as of 21 June (cut-off date for deaths associated with infections on or before 31 May), similar to estimates from a different retrospective study [10].

To further dissect the relationships between model parameters and estimated quantities we next turn to correlation analysis. The following notable pairs of epidemiologically and statistically significant variables were found: $I_0$ versus date of first reported case (after 21 January) with Spearman correlation $\varrho = 0.32$ ($p = 1.7 \times 10^{-2}$); mortality (% population) versus $\psi$ with $\varrho = -0.67$ ($p = 1.4 \times 10^{-8}$); true case estimate versus $\psi$ with $\varrho = -0.85$ ($p = 2.2 \times 10^{-16}$); $\psi$ versus half-life fatigue with $\varrho = -0.52$ ($p = 4.2 \times 10^{-5}$); peak daily cases versus $\psi$ with $\varrho = -0.89$ ($p = 8.5 \times 10^{-20}$); $\mathcal{R}_0$ versus $\psi$ with $\varrho = 0.43$ ($p = 7.3 \times 10^{-3}$); time to peak daily cases (relative to first reported case in USA) $\varrho = -0.62$ ($p = 2.7 \times 10^{-7}$); cumulative reported cases versus IFR with $\varrho = 0.46$ ($p = 4.0 \times 10^{-4}$); cumulative case estimate/reported cases versus IFR with $\varrho = -0.60$ ($p = 1.3 \times 10^{-6}$). This analysis (see figure 5; electronic supplementary material, figure S18 for all correlation plots, as well as electronic supplementary material, figure S19 for correlation values between all parameters and calculated quantities) together with our three variable scatter plots in figure 4 suggest that $\psi$ dominates fatigue half-life and $\mathcal{R}_0$ in terms of statistically significant correlation between model parameters and quantities of interest such as cumulative and peak daily case totals, daily cases on 31 May 2020, and half-life fatigue.

Furthermore, we observe that $\psi$ and estimated true case totals (shown in figure 5$c$) as well as peak daily case totals (($d$) of the same figure) follow an inverse proportionality relationship with respect to $\psi$. Indeed, as derived in [1], for the case $\alpha = 0$ (indefinite quarantine period), the final cumulative infected, $\mathcal{C} = \int_0^\infty \lambda(t)S(t)dt$ and peak infected, $\mathcal{P} = \max \frac{I(t)}{N}$, (as % of population) of model (2.1) satisfy

$$\ln(s_\infty) = \mathcal{R}_0(s_\infty - 1), \quad \mathcal{C} = \frac{1}{1 + \psi}(1 - s_\infty) \tag{4.1}$$

and

$$\mathcal{P} = \frac{1}{1 + \psi}\frac{1}{\mathcal{R}_0}(\mathcal{R}_0 - \ln \mathcal{R}_0 - 1), \tag{4.2}$$

where $s_\infty = \lim_{t \to \infty} S(t)/N$ and $I_0 \approx 0$ (at start of outbreak). In each formula, the factor $1/(1 + \psi)$ is multiplied by the corresponding classical relations for final and peak outbreak size. In view of our model fitting, the true cumulative and peak case estimates of each state fall roughly into an inverse proportionality with $\psi$, although states with low fatigue half-life ($\alpha$ much larger than zero) stray from this pattern, as shown in figure 6. In particular, by simply calibrating the inverse proportionality relation, $\mathcal{C} = K_{NY}/(1 + \psi)$, to the estimated parameters and cumulative cases of New York (which has small $\psi$ and $\alpha \approx 0$), we approximately fit the rest of the state values, with the analogous result for peak cases (see figure 6). The different values of $\mathcal{R}_0$ and $\alpha$ in each state, however, shift the respective case numbers from the expected values predicted by inverse relation with $\psi$ (figure 6). Thus while $\psi$

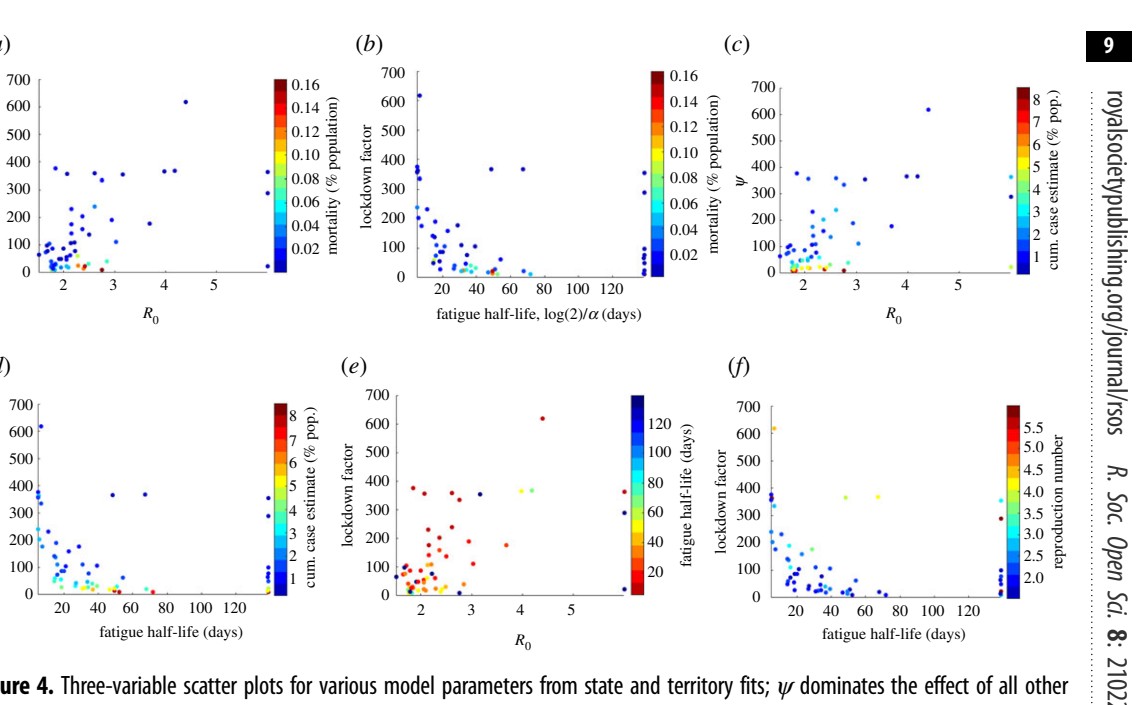

**Figure 4.** Three-variable scatter plots for various model parameters from state and territory fits; $\psi$ dominates the effect of all other model parameters; of particular note is that high $\psi$ values tend to be associated with low fatigue half-life (high $\alpha$) values.

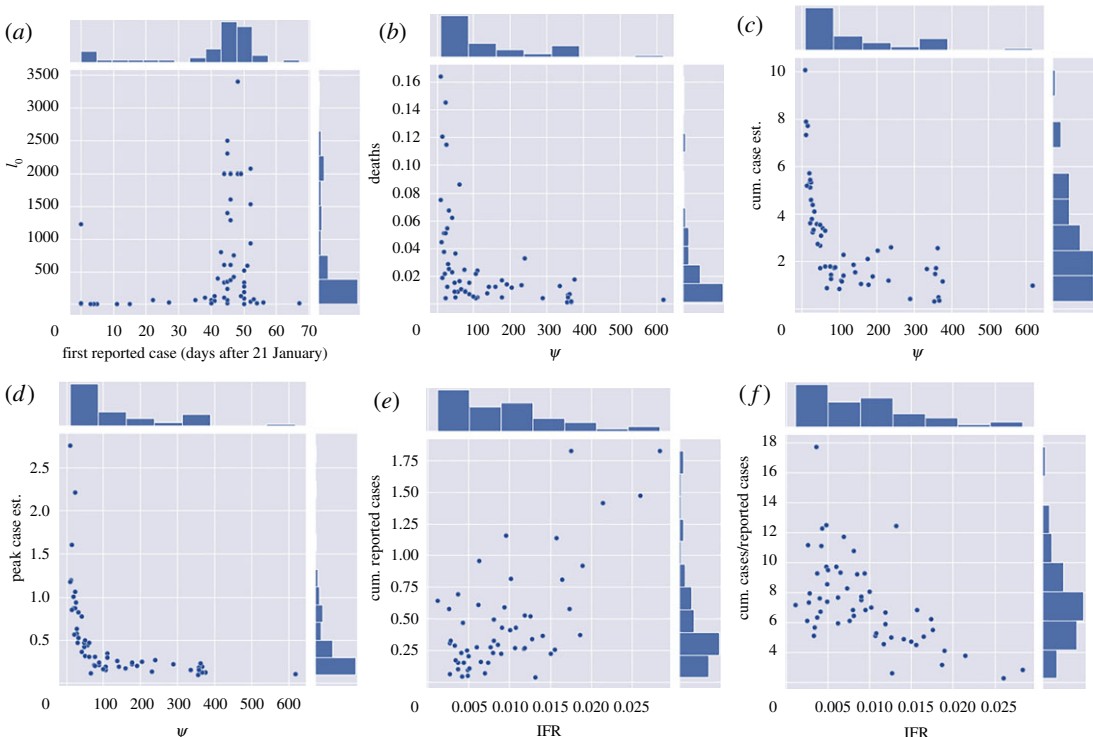

**Figure 5.** For correlation analysis involving $\alpha$ North Dakota was not included because the fit value was not practically identifiable (table 2). (a) $I_0$ versus date of first reported case (days after 21 January). (b) Mortality (% pop) versus $\psi$. (c) True case estimate (% pop) versus $\psi$. (d) Peak daily cases (% pop) versus $\psi$. (e) Cumulative reported cases versus IFR. (f) Cumulative case estimate/reported cases versus IFR. See electronic supplementary material, figure S19 for all correlation values, and Results section for highlighted values.

has strongest impact (figures 4–6), with rapidly escalating costs as the lockdown (rate) factor decreases below a critical value, $\mathcal{R}_0$ and $\alpha$ still have an influence on outcome (figures 9 and 10). We will examine these relationships further with counterfactual simulation studies (figure 9).

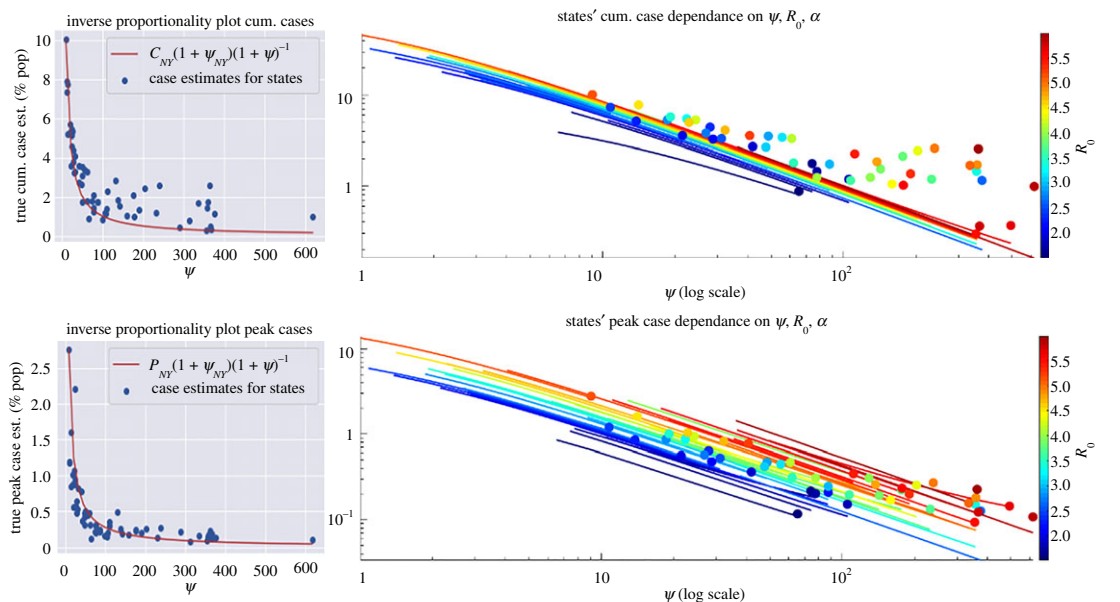

**Figure 6.** Both cumulative cases and $\psi$ (top right) and peak cases (bottom right) and $\psi$ follow the inverse proportionality relationship described in [1]. The plotted lines indicate change in cumulative (peak) case estimate as $\psi$ is varied from 10% of its fit value up to the fit value itself. The colour gradient of the lines from dark blue to dark red is indicative of increasing reproduction number $\mathcal{R}_0$. This suggests that while $\psi$ dominates these relationships, as our correlation analysis also indicates, $\mathcal{R}_0$ still has an impact on outcome with regard to cumulative and peak case total. Top left shows the inverse proportionality relationship for the New York parameter set going from fit $\psi$ to the maximum observed value; bottom left is the same except for peak daily cases where the red lines follow equations (4.1) and (4.2), respectively.

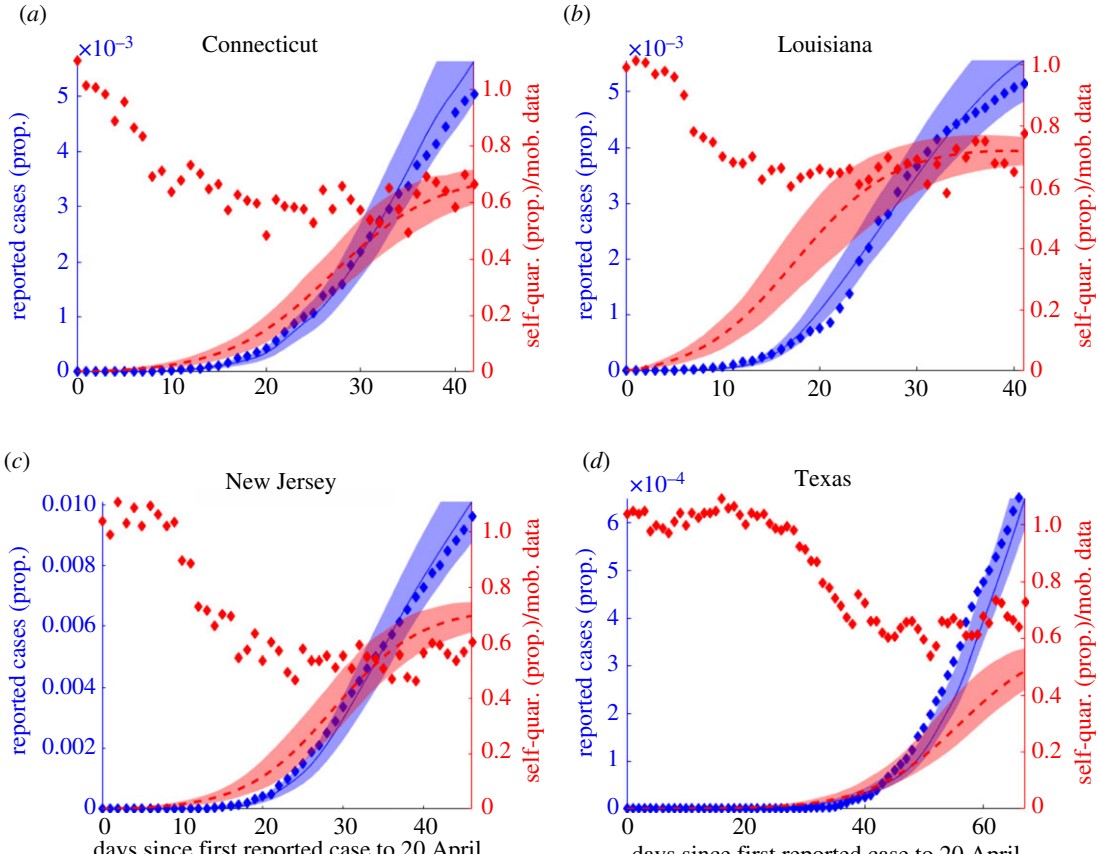

**Figure 7.** Model Lockdown factor, $\psi$ captures reduction in mobility, similar to that suggested by mobility data from [11], well across epidemic trajectories during time period for which it is available (to 20 April 2020). Diamonds represent data, lines model fit, shaded regions confidence intervals.

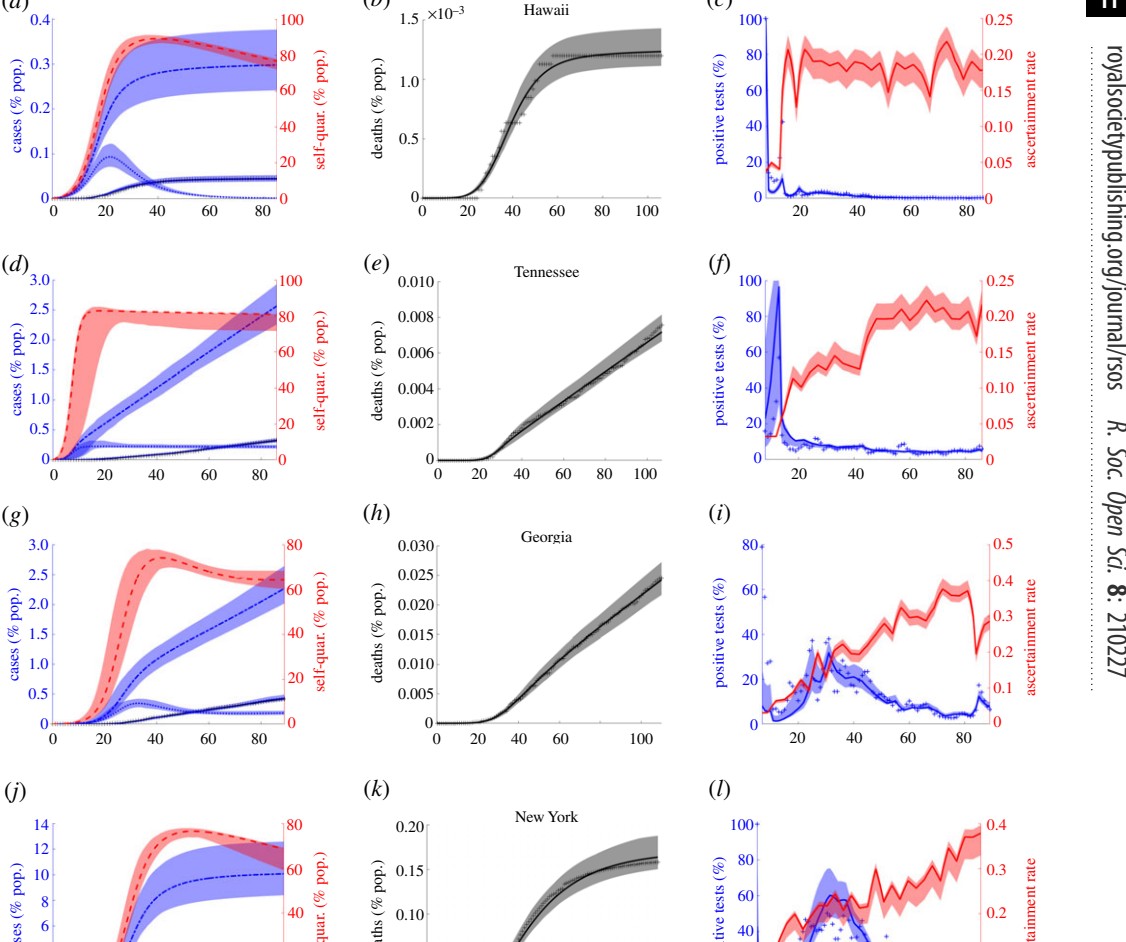

**Figure 8.** Representative examples of different relative value combinations of $\psi$ and $\alpha$, quality of strategy decreases from top to bottom row. In $(a),(d),(g),(j)$, the solid blue line represents the reported case fit, the dashed and dotted line represents cumulative case estimate, and the dotted line represents daily case estimate.

Also of note is the suggested relationships between $\mathcal{R}_0$ and $I_0$ as well as $\mathcal{R}_0$ and time to peak daily case incidence. Figure 5a shows the aforementioned positive correlation between estimated $I_0$ and date of first reported case (in terms of days after 21 January). This is further supported by the plots of time variable $\mathcal{R}_e$ in figure 2a where it can be seen that, in general, states which have a later date for their first reported case tend to have higher reproduction number (for specific values see table 2). Two relationships of note are the positive relationship between IFR and reported case estimate and the negative relationship between IFR and ratio of true to reported cases. The positive relationship between IFR and reported case estimate and the negative relationship between IFR and ratio of true to reported cases probably explain each other: as more cases and deaths were recorded greater effort was put into case reporting. And even in locales with successful outbreak responses such as Finland [7], or Hawaii in our own model (figure 8), we note that the proportion of cases undetected in the early days of the outbreak is very high. For our US fit, we estimated an IFR of 0.012 (95% confidence interval 0.0108 to 0.0133), which is in line with the observed IFR for countries with very high levels of testing such as South Korea [25] and Finland [7].

Correlation analysis alone does not give the full picture. To further examine and differentiate the effects of $\alpha$, $\psi$ and $\mathcal{R}_0$ on epidemic trajectories and suggest potential alternative strategies for management of subsequent waves of COVID-19 we carried out both counterfactual simulation studies and sensitivity analysis on these key model parameters (figures 9–11; electronic supplementary

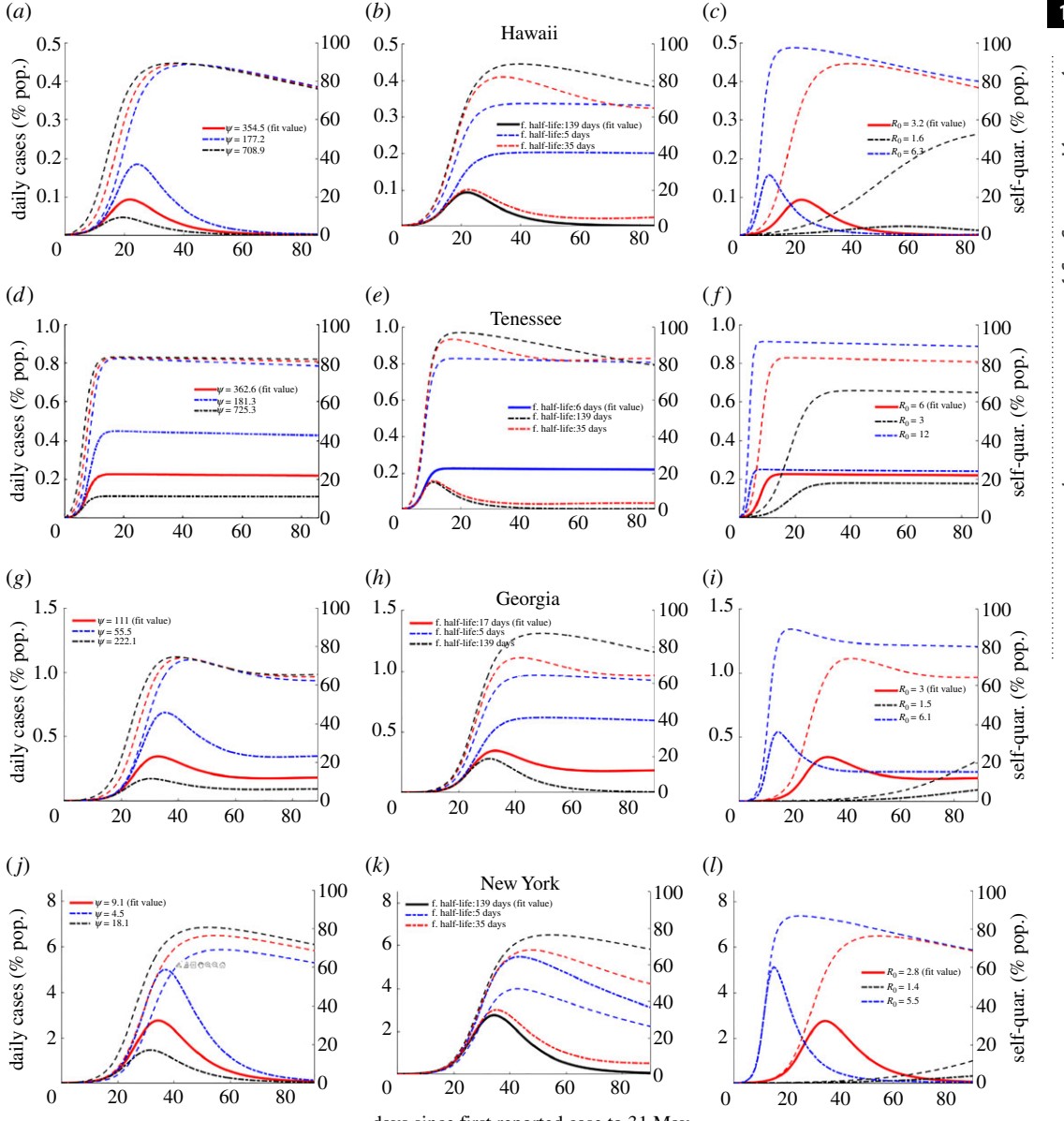

**Figure 9.** The differing effects of relative magnitudes and variation of $\psi$, $\alpha$ and $\mathcal{R}_0$ on epidemic trajectory as demonstrated by sample states shown in figure 8 and the United States fit 3. For specific fit parameter values, see table 2. Column 1 shows variation in $\psi$, column 2 in $\alpha$ and column 3 in $\mathcal{R}_0$. Dashed lines indicate (self-)quarantined individuals over time corresponding to epidemic trajectory of the same colour. Black indicates best-case scenario, red intermediate, blue worst, bold line is fit value, dashed and dotted are alternative.

material, figure S16). This was done using the fit model parameters for selected states which broadly speaking represent the range of outbreak responses which occurred in the USA. These are, in order of increasing cumulative outbreak size as of 31 May (figure 8):

  (i) rapid lockdown with high half-life (RLHH) as represented by Hawaii;
  (ii) rapid lockdown with low half-life (RLLH) as represented by Tennessee;
  (iii) intermediate lockdown with intermediate half-life (ILIH) as represented by Georgia;
  (iv) slow lockdown with high half-life (SLHH) as represented by New York.

As can be seen in these figures, the states all reported their first confirmed case within a week of one another, but had significantly different outcomes. Taken as per cent of population, Hawaii had both the lowest cumulative case total across all states and territories and one of the briefest outbreaks in terms of

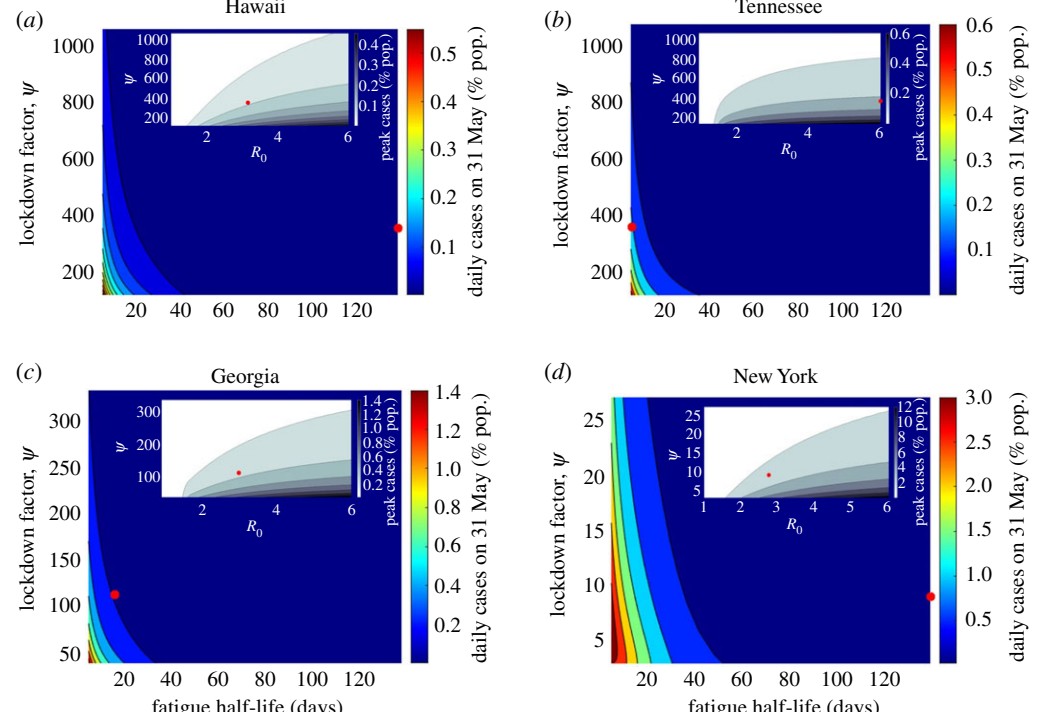

**Figure 10.** Daily cases on 31 May 2020 and peak daily cases (see insert) as functions of $\psi$, lockdown fatigue half-life and $\psi$, $\mathcal{R}_0$, respectively, magnitude of change is primarily determined by $\psi$ as can be seen by the scales of the y-axes. $\alpha$ determines duration of outbreak, with critical half-life threshold of 30–60 days. Points indicate fit parameter values.

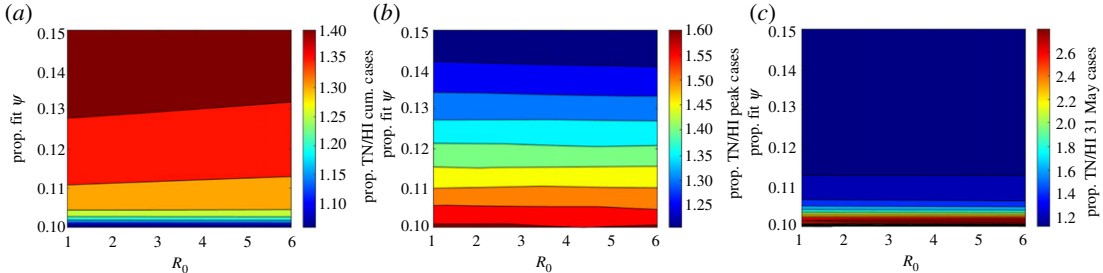

**Figure 11.** Ratio of Tennessee to Hawaii (*a*) cumulative cases; (*b*) peak cases; (*c*) daily cases on 31 May as $\psi$ is varied from its 15% to 10% its fit value. Given similar fit $\psi$ values difference in peak and daily cases is attributable to differing half-life fatigue with magnitude determined by change in $\psi$, whereas ratio of cumulative cases approaching one reflects the inverse proportionality relationship between $\psi$ and cumulative cases (figure 6, equation (4.2), [1]).

increasing daily case totals. All other considerations aside response RLHH represents the optimal strategy in terms of both peak and cumulative daily case totals (see figure 8 and table 2; electronic supplementary material, figures S1–S14).

Tennessee highlights the importance of fatigue half-life. In the baseline fit epidemic trajectory, despite having a lockdown factor similar to Hawaii, 362.6 versus 354.5 (CIs 188.0–408.1 versus 274.4–432.1), Tennessee's cumulative case totals are an more than eight times (2.57%, CI 2.27–2.86% versus 0.30% CI 0.25–0.39%) as high as a result of significant linear growth in daily cases not far off the peak value. While this response could potentially keep hospitals from being overwhelmed provided sufficiently high $\psi$ (see figure 8), it both prolongs the outbreak and leads to more cases than response RLHH (see figures 8, 10, 11; electronic supplementary material, figure S16).

Georgia represents a number of states for which both the fit $\psi$ and fatigue half-life values fall into a middle ground between the maximal and minimal fit values across all states and territories, 111.0 (CI

84.9–153.6) and 16.3 (CI (11.2–20.8)) days, respectively. As can be seen in the fit trajectory (figure 8 and table 2) despite having a $\psi$ value much lower than that of Tennessee and similar dates of first reported case, the two states have relatively similar cumulative case totals, (CI 2.05–2.57%) for Georgia, suggesting that there may be a critical threshold for fatigue half-life which is examined further below (figures 9–11).

New York is representative of the final observed response, slow lockdown with high fatigue half-life (CIs 7.0–11.3 and 77.5–138.6 days). Despite having both the highest reported cumulative and estimated true case totals across all states and territories New York's outbreak was essentially over at a time when many states were experiencing significant daily case totals despite similar dates of first reported case (see figure 8; electronic supplementary material, figures S1–S14).

Our results suggest that the optimal response strategy in terms of case totals, and therefore also fatalities, remains rapid lockdown with high fatigue half-life (response RLHH) (see figures 4, 8 and 9 and table 2; electronic supplementary material, figures S1–S14). Here, the speed of lockdown is the critical factor as our model considers that individuals and/or government will eventually react to escalating cases, reaching similar numbers of self-quarantined (distinct for each state parameter set) in the counter-factual simulations of varying $\psi$ with the difference being delayed lockdown resulting in order of magnitude more cases. While no substitute for the rapid lockdown strategy, the next best intervention may be sustained public social distancing and mask wearing, targeting transmission reduction rather than removing susceptibles all together, to reduce $\mathcal{R}_0$, see figures 9 and 10. This has the added benefit of reducing the number of individuals that need to be quarantined, with those that are able to self-quarantine doing so preferentially throughout the entire outbreak, but with at least half the quarantined population not returning to normalcy for a period of 30 days (figure 10). Indeed, across all four example states, and regardless of model parameters, self-quarantine periods lasting longer than this threshold greatly reduce the daily case total on 31 May, suggesting that if this is achieved, other methods of managing the outbreak, such as contact tracing, could then be employed to manage subsequent cases.

Sufficiently low fatigue half-life can lead to sustained linear growth in cumulative cases (or sustained steady state of daily cases) regardless of $\psi$ value (figures 9–11). Despite Tennessee and Hawaii having similar fit $\psi$ values, the daily case load on 31 May and the peak daily case incidence for Tennessee (as percentage of state population) are approximately 300% and 160% greater than those of Hawaii respectively when $\psi$ is reduced by 90%, suggesting that this is attributable to their different $\alpha$ values (figure 11), which fall on opposite ends of the considered range. Cumulative cases in contrast become approximately equal, highlighting the inverse proportional relationship between cumulative cases and $\psi$ (figure 6, equation (4.2), [1]). Thus our model suggests that lockdown fatigue, $\alpha$, primarily affects whether or not daily case incidence on 31 May is close to zero relative to peak cases, where lockdown factor (or speed), $\psi$, modulates the peak level. A similar result is suggested with respect to $\mathcal{R}_0$ and peak daily case total as well as date of peak, importantly with low fatigue half-life significantly dampening the impact of lowered $\mathcal{R}_0$ (see figures 9 and 10). While caution must be exercised with short-duration or high-turnover lockdowns (responsive to accumulating cases), governments and individuals are also reluctant to return to strict lockdown. Given this reality, while response RLHH has the best outcomes, it is important to consider the viability of alternative strategies which might be implemented for future outbreaks.

## 5. Discussion

In this study, we fit a unified model to case, death and testing data from all 50 states as well as Washington DC and four outlying territories in order to quantify impacts of a wide variety of lockdown entry/exit responses, which we correlate with true case estimates for insight into the dynamic COVID-19 outbreak control in the USA. Despite the heterogeneous nature of exhibited responses to the outbreak within each state, we were able to obtain quality fits for all considered states and territories. A crucial step in doing so was incorporating testing data, $\tau(t)$, by calibrating the ascertainment rate, $\rho(\tau(t))$, as a saturating function of *per capita* tests, which provided extra confidence in our fitting because daily per cent positive tests were mimicked in each state (see figures 3 and 8; electronic supplementary material, figures S1–S14). The inferred true cumulative case totals demonstrate high varying levels of under-reporting, with an estimated 5.85 (CI 5.31–6.30) cases per reported cases (cumulative) overall in the USA up to 31 May 2020. Furthermore, our modelling framework allows us to infer infection fatality ratio, found to be 1.2% for the USA over the time period considered (CI 1.08–1.33%). These are probably only estimates for (reported and unreported)

symptomatic cases which do not account for a significant extent of asymptomatic individuals, since testing, reported case and mortality data are all derived mostly from symptomatic infections.

Our model captured increasing case ascertainment rates through time (as testing increased), with a significant positive correlation to the amount of cases reported in a state (figure 5; electronic supplementary material, figures S18 and S19), in contrast to a recent study in France showing a negative relationship between ascertainment ratio and reported cases as larger case counts hampered their substantial contact tracing programme [7]. In the USA, comparatively little effort was put into contact tracing, and the particular geographical or political features of COVID-19 spread, along with higher case burden focusing community attention on the disease, may have contributed to the reverse correlation. Faced with a spreading outbreak, high numbers of undetected infection, and no nationwide control policy, the USA adopted heterogeneous responses centred around some form of lockdown, which we characterize according to parameter fits of quarantine entry/exit rate for each state (table 2).

Our model fitting results suggest, broadly speaking, that the distinct responses can be divided into one of four categories (see figure 8 and table 2; electronic supplementary material) (arranged in order of generally increasing cumulative case totals): RLHH, RLLH, ILIH and SLHH. Using correlation analysis of the key parameters (figures 2 and 5; electronic supplementary material, figures S18 and S19), and outcomes over all US states, along with counterfactual simulation studies and sensitivity analysis (figures 9–11, electronic supplementary material, figure S16) for representative states of these range of responses (Hawaii, Tennessee, Georgia and New York), we translate the unified model fits into assessing the variable aspects of reactive lockdowns or self-quarantine (speed, scale, duration/fatigue) (figures 6 and 9–11; electronic supplementary material, figure S16) in combination with sustained interventions aimed at reducing $\mathcal{R}_0$, such as public face mask wearing, social distancing and contact tracing.

Lockdown (rate) factor $\psi$ played the largest role in outbreak severity (figures 4 and 9–11; electronic supplementary material, figure S16), in particular both cumulative and peak cases roughly followed an inverse proportionality relationship with $\psi$ (figure 6), which was theoretically derived and applied to the epidemic in China [1] in the instance self-quarantine (lockdown) exit rate, $\alpha$, is zero. Differences in $\mathcal{R}_0$ also influenced outbreak shape, where its reduction mainly results in both a delayed peak in daily case totals and decreased size of peak daily case incidence, along with lessening the scale of lockdown needed to control the disease (figures 9–11; electronic supplementary material, figure S16). Low fatigue half-life ($\alpha$ significantly larger than zero) in some US states did impact their outbreaks though, the daily case load at the end of the time-frame considered (31 May) being large, with magnitude determined primarily by $\psi$, but also $\mathcal{R}_0$ (figures 8–10). Importantly, independent of all other model parameters our model suggests a critical threshold of 30–60 days for half-life return to normalcy, that is for 50% of the quarantined population to exit lockdown (see figures 10 and 11; electronic supplementary material, figure S16). With this threshold indicating the approximate point at which there is significant reduction in daily case load within approximately 90 days of first reported case, perhaps allowing for other less invasive control methods, such as contact tracing, to be implemented.

Our results provide insights for management of future outbreaks. The optimal strategy purely in terms of cumulative and peak daily cases, and therefore also fatalities, remains response RLHH (rapid lockdown with high half-life) (figures 9–11; electronic supplementary material, figure S16). However, in order to balance economic/social concerns in addition to the purely epidemiological, we suggest a new response to an outbreak compared with those observed in the USA first wave: rapid measured lockdown with intermediate half-life. This strategy describes implementation of rapid reactive lockdown as soon as possible in conjunction with subsequent wave being detected, lasting at least 30 days before 50% return to normalcy, with the underlying $\mathcal{R}_0$ and public adherence to proper social distancing and mask wearing determining the scale of closures during lockdown (where at-risk or sectors responsible for large-scale spread are preferentially included in quarantine).

Both $\psi$ and $\alpha$ (or lockdown fatigue) are measures of social mobility responses influenced by public perception of risk [12], with $\psi$ tied to disease incidence (figure 7). By contrast, $\alpha$ is not directly assumed dependent upon force of infection, but rather represents either a perception that the disease has been brought under control or individual/government fatigue from lockdown after corresponding amount of time characterized by the population's quarantine half-life. Thus if public health officials wish to use them as levers to control an outbreak; consideration must be given to how these factors can be effected by the actions of government officials. A study of multiple proxies of social distancing and government actions, at both local and state levels, [12], found that early information focused efforts and emergency declarations had measurable effect on individual behaviour. Thus we propose that when testing results indicate the potentiality for a subsequent wave of COVID-19 that public health officials rapidly disseminate information about both the need for a return to self-quarantine measures and for individuals to continue

such measures for at least 30 days, with emphasis on return to normalcy not at the first sign of reduced spread, but rather after a period of sustained reduction in spread.

The quick response and critical duration of quarantined sectors will perhaps allow case numbers to be sufficiently reduced after the 30-day period for contact tracing to be feasible (figures 9 and 10) and in combination with broader measures aimed at lowering $\mathcal{R}_0$ (e.g. face masks) can potentially prevent any substantial subsequent wave until effective vaccines are widely taken by the population (figure 9; electronic supplementary material, figure S16; [1]).

Data accessibility. All relevant code https://github.com/jcmacdonald-codesData/programfilesUSCOVID and data reshaped to make best use of Matlab data structures, as well as data such as fit parameters generated by the fitting process are available at https://github.com/jcmacdonald-codesData/DataFilesUSCOVID. These have been archived together at the Zenodo repository: doi:10.5281/zenodo.4891838. Raw data were taken from the following locations: where shown, indicated stay-at-home orders are based upon CDC reports [23]. Population levels are from US Census Bureau 2019 projections [20]. To obtain parameter estimates for our model in each state and territory, we used cumulative death and case totals [24] as well as daily testing data, inferred from [21]. Mobility data, used in figure 7 is from [11].

Competing interests. We declare we have no competing interests.

Funding. J.C.M., C.B. and H.G. are supported by a U.S. National Science Foundation RAPID grant (no. DMS-2028728). H.G. was also supported by an NSF grant (no. DMS-1951759) and a grant from the Simons Foundation/SFARI (638193). C.B. is partially supported by an NSF grant (no. DMS-1815095).

Acknowledgements. The authors would like to thank Fadoua Yahia (Department of Mathematics, University of Louisiana at Lafayette) for her contributions to an earlier version of this paper.

# Appendix A

(See tables 2–5)

**Table 2.** Key parameter values and inferred quantities. True case estimate is as of May 31, 2020. All parameters are practically identifiable except $\alpha$ for North Dakota.

| State | $\mathcal{R}_0$ | $\psi$ | F. half-life (days) | IFR | $I_0$ | cum. case est. (%) | cum. case/ rep. case | D. cases 31 May (%) |
|---|---|---|---|---|---|---|---|---|
| Alabama | 2.15 | 176.0 | 7.5 | 0.008 | 1538.0 | 2.11 | 6.44 | 0.20 |
| Alaska | 3.98 | 365.8 | 48.5 | 0.003 | 588.6 | 0.48 | 7.34 | 0.02 |
| Arizona | 1.85 | 376.1 | 4.9 | 0.016 | 6.1 | 1.15 | 4.47 | 0.13 |
| Arkansas | 2.61 | 358.9 | 4.9 | 0.004 | 186.0 | 1.72 | 7.57 | 0.18 |
| California | 2.15 | 231.3 | 11.0 | 0.012 | 4.9 | 1.19 | 4.54 | 0.11 |
| Colorado | 1.77 | 28.7 | 50.9 | 0.009 | 2000.0 | 3.23 | 7.49 | 0.14 |
| Connecticut | 2.39 | 14.1 | 49.4 | 0.016 | 747.4 | 7.72 | 6.80 | 0.23 |
| Delaware | 1.82 | 10.8 | 71.9 | 0.006 | 523.0 | 7.33 | 7.63 | 0.36 |
| District of Columbia | 1.77 | 10.7 | 52.3 | 0.010 | 606.1 | 7.88 | 6.81 | 0.35 |
| Florida | 2.94 | 190.0 | 15.6 | 0.011 | 65.3 | 1.36 | 5.12 | 0.11 |
| Georgia | 3.03 | 111.0 | 16.3 | 0.011 | 51.6 | 2.27 | 5.26 | 0.18 |
| Guam | 6.00 | 23.6 | 138.6 | 0.001 | 17.3 | 4.58 | 7.17 | 0.03 |
| Hawaii | 3.16 | 354.5 | 138.6 | 0.004 | 17.7 | 0.30 | 6.68 | 0.00 |
| Idaho | 2.21 | 110.2 | 30.9 | 0.004 | 2075.0 | 1.39 | 9.29 | 0.06 |
| Illinois | 1.85 | 26.8 | 34.0 | 0.019 | 11.5 | 3.78 | 4.11 | 0.28 |
| Indiana | 1.94 | 18.6 | 46.7 | 0.008 | 1392.4 | 5.30 | 10.77 | 0.24 |
| Iowa | 1.75 | 21.5 | 67.8 | 0.006 | 419.8 | 3.59 | 5.93 | 0.27 |
| Kansas | 2.05 | 55.7 | 18.3 | 0.003 | 2000.0 | 3.39 | 11.17 | 0.23 |

(Continued.)

| State | $\mathcal{R}_0$ | $\psi$ | F. half-life (days) | IFR | $I_0$ | cum. case est. (%) | cum. case/ rep. case | D. cases 31 May (%) |
|---|---|---|---|---|---|---|---|---|
| Kentucky | 2.36 | 158.8 | 22.8 | 0.015 | 61.2 | 1.05 | 4.73 | 0.07 |
| Louisiana | 2.49 | 32.2 | 39.4 | 0.016 | 2000.0 | 4.08 | 5.05 | 0.15 |
| Maine | 2.49 | 137.7 | 15.2 | 0.004 | 585.6 | 1.86 | 12.24 | 0.14 |
| Maryland | 2.10 | 19.1 | 36.9 | 0.010 | 329.6 | 5.70 | 7.00 | 0.32 |
| Massachusetts | 2.29 | 24.4 | 32.1 | 0.021 | 2.5 | 5.34 | 3.77 | 0.26 |
| Michigan | 2.84 | 40.6 | 36.6 | 0.017 | 1999.9 | 3.57 | 6.20 | 0.15 |
| Minnesota | 1.84 | 31.1 | 26.7 | 0.010 | 237.2 | 3.31 | 8.02 | 0.34 |
| Mississippi | 2.61 | 239.2 | 4.9 | 0.013 | 330.1 | 2.60 | 5.01 | 0.27 |
| Missouri | 1.79 | 87.3 | 18.7 | 0.009 | 1293.1 | 1.73 | 7.71 | 0.13 |
| Montana | 4.18 | 366.7 | 67.3 | 0.005 | 45.1 | 0.36 | 7.37 | 0.01 |
| Nebraska | 1.81 | 27.9 | 18.6 | 0.004 | 24.3 | 4.40 | 6.34 | 0.46 |
| Nevada | 2.05 | 47.6 | 33.6 | 0.006 | 600.7 | 2.66 | 9.72 | 0.12 |
| New Hampshire | 1.65 | 74.2 | 16.1 | 0.014 | 122.1 | 1.77 | 4.85 | 0.18 |
| New Jersey | 2.42 | 22.3 | 49.5 | 0.028 | 800.7 | 5.13 | 2.81 | 0.15 |
| New Mexico | 1.72 | 105.2 | 21.2 | 0.019 | 268.8 | 1.18 | 3.16 | 0.10 |
| New York | 2.75 | 9.1 | 138.6 | 0.018 | 1224.5 | 10.06 | 5.51 | 0.07 |
| North Carolina | 2.36 | 203.3 | 5.7 | 0.005 | 391.0 | 2.45 | 9.72 | 0.25 |
| North Dakota | 1.51 | 65.4 | 138.5 | 0.013 | 94.8 | 0.87 | 2.60 | 0.07 |
| N. Mariana Islands | 6.00 | 288.8 | 138.6 | 0.013 | 20.0 | 0.42 | 12.40 | 0.00 |
| Ohio | 1.75 | 41.9 | 27.0 | 0.009 | 3400.0 | 2.72 | 9.20 | 0.19 |
| Oklahoma | 1.68 | 77.4 | 34.3 | 0.007 | 2300.7 | 1.45 | 9.33 | 0.06 |
| Oregon | 2.14 | 106.6 | 39.3 | 0.005 | 108.1 | 1.16 | 12.47 | 0.05 |
| Pennsylvania | 2.06 | 22.4 | 30.9 | 0.009 | 2492.9 | 5.44 | 9.25 | 0.29 |
| Puerto Rico | 1.91 | 49.2 | 138.6 | 0.004 | 935.9 | 1.72 | 17.70 | 0.04 |
| Rhode Island | 2.28 | 60.8 | 15.0 | 0.026 | 15.0 | 3.30 | 2.24 | 0.29 |
| South Carolina | 2.16 | 142.2 | 14.3 | 0.008 | 342.0 | 1.55 | 6.80 | 0.13 |
| South Dakota | 1.95 | 48.6 | 14.6 | 0.003 | 125.3 | 3.53 | 6.09 | 0.33 |
| Tennessee | 6.00 | 362.6 | 5.3 | 0.003 | 108.0 | 2.57 | 7.94 | 0.22 |
| Texas | 2.00 | 87.5 | 20.0 | 0.005 | 72.7 | 1.75 | 8.54 | 0.14 |
| Utah | 2.07 | 356.7 | 4.9 | 0.003 | 61.5 | 1.45 | 5.07 | 0.15 |
| Vermont | 2.22 | 77.5 | 138.6 | 0.007 | 372.2 | 1.25 | 8.28 | 0.01 |
| Virgin Islands | 1.67 | 98.8 | 138.6 | 0.007 | 74.0 | 0.83 | 11.72 | 0.01 |
| Virginia | 1.79 | 13.8 | 138.6 | 0.004 | 1599.3 | 5.18 | 11.10 | 0.29 |
| Washington | 2.12 | 62.1 | 54.3 | 0.012 | 2.9 | 1.79 | 6.63 | 0.05 |
| West Virginia | 3.69 | 177.1 | 28.9 | 0.005 | 24.3 | 1.01 | 9.48 | 0.07 |
| Wisconsin | 2.75 | 334.6 | 6.6 | 0.008 | 1.0 | 1.69 | 6.09 | 0.15 |
| Wyoming | 4.40 | 617.8 | 6.6 | 0.003 | 1.0 | 0.98 | 5.67 | 0.10 |
| United States | 2.33 | 49.7 | 27.9 | 0.012 | 20.3 | 3.07 | 5.85 | 0.16 |
| mean | 2.50 | 131.0 | 17.6 | 0.009 | 627.1 | 2.82 | 7.30 | 0.16 |
| median | 2.15 | 77.4 | 30.9 | 0.008 | 268.8 | 2.11 | 6.81 | 0.15 |
| std. dev. | 1.59 | 312.4 | 78.4 | 0.003 | 1086.8 | 0.80 | 0.54 | 0.07 |

**Table 3.** Approximate 95% confidence intervals from uncertainty quantification. All parameters except $\alpha$ for North Dakota are practically identifiable at 70% noise level.

| State | $\mathcal{R}_0$ | $\psi$ | F. half-life (days) | IFR | $I_0$ | true case est. | T. case/r. case | D. cases 31 May (%) |
|---|---|---|---|---|---|---|---|---|
| Alabama | (1.59,2.78) | (61.4,286.8) | (4.9,18.4) | (0.007,0.01) | (768.98,2306.93) | (1.83,2.44) | (5.73,6.81) | (0.152,0.238) |
| Alaska | (2.88,5.56) | (405.9,621.1) | (35.2,58.8) | (0.003,0.005) | (232.85,698.56) | (0.32,0.43) | (5.32,6.31) | (0.014,0.02) |
| Arizona | (1.73,1.94) | (124.3,414.8) | (4.9,17.3) | (0.014,0.019) | (3.06,9.18) | (1.01,1.29) | (4.02,4.77) | (0.101,0.15) |
| Arkansas | (2.12,3.24) | (209.9,404.5) | (4.9,8.3) | (0.003,0.005) | (97.05,279.07) | (1.48,1.96) | (6.67,7.97) | (0.154,0.203) |
| California | (2.05,2.31) | (144.1,485) | (4.9,18.5) | (0.01,0.014) | (2.47,7.42) | (1.06,1.4) | (4.03,4.89) | (0.088,0.125) |
| Colorado | (1.58,2.00) | (20.6,30.5) | (28.5,138.6) | (0.008,0.011) | (1000.00,3000.00) | (2.84,3.75) | (6.6,8.32) | (0.098,0.187) |
| Connecticut | (2.17,2.64) | (11.5,16.5) | (37.5,100.8) | (0.014,0.019) | (373.69,1121.06) | (6.77,8.97) | (5.98,7.48) | (0.13,0.33) |
| Delaware | (1.68,2.06) | (8.7,15) | (28.2,138.6) | (0.005,0.008) | (261.52,784.57) | (6.48,8.95) | (6.82,8.51) | (0.278,0.486) |
| Dist. of Columbia | (1.64,2.02) | (7.7,15.2) | (27.8,138.6) | (0.008,0.012) | (303.07,898.06) | (7.04,9.24) | (6.2,7.59) | (0.264,0.475) |
| Florida | (2.64,3.31) | (138,248) | (11.8,22.4) | (0.009,0.013) | (32.67,98) | (1.23,1.59) | (4.6,5.4) | (0.09,0.131) |
| Georgia | (2.76,3.42) | (84.9,153.6) | (11.2,20.8) | (0.01,0.014) | (25.79,77.38) | (2.05,2.57) | (4.82,5.7) | (0.159,0.222) |
| Guam | (5.32,6) | (20.3,29.4) | (99,138.6) | (0.001,0.001) | (13.89,25.97) | (3.78,5.28) | (6.23,8.03) | (0.026,0.071) |
| Hawaii | (2.88,3.64) | (274.3,432.1) | (100.9,138.6) | (0.003,0.006) | (8.86,26.59) | (0.25,0.39) | (6.03,7.53) | (0.001,0.002) |
| Idaho | (2.82,4.13) | (108.1,158.1) | (33.2,45.2) | (0.003,0.005) | (228.01,684.02) | (1.05,1.46) | (7.58,9.28) | (0.046,0.076) |
| Illinois | (1.78,1.95) | (19.5,38.8) | (17.7,138.2) | (0.014,0.021) | (5.73,17.18) | (3.2,4.57) | (3.66,4.58) | (0.2,0.362) |
| Indiana | (1.77,2.14) | (14,23.5) | (26.3,138.6) | (0.007,0.011) | (709.39,2088.63) | (4.43,6.22) | (9.38,12.21) | (0.16,0.356) |
| Iowa | (1.64,1.96) | (16.5,33.3) | (22.1,138.6) | (0.005,0.007) | (209.91,625.94) | (3.2,4.18) | (5.52,6.5) | (0.222,0.356) |
| Kansas | (1.75,2.56) | (42.9,69.3) | (14.1,23.8) | (0.002,0.003) | (1000,3000) | (2.95,4) | (9.96,12.47) | (0.176,0.283) |
| Kentucky | (2.16,2.68) | (122.2,208) | (16,32) | (0.012,0.017) | (306,91.8) | (0.93,1.22) | (4.23,5.11) | (0.055,0.091) |
| Louisiana | (2.23,2.88) | (26.4,38.6) | (33.2,51.9) | (0.014,0.02) | (1000,3000) | (3.56,4.69) | (4.59,5.5) | (0.101,0.204) |
| Maine | (2.06,3.19) | (103.3,188.2) | (11.6,19.7) | (0.004,0.006) | (292.82,878.45) | (1.56,2.19) | (10.97,13.46) | (0.11,0.177) |
| Maryland | (1.96,2.29) | (15.1,24.8) | (22.9,93) | (0.009,0.013) | (164.79,494.36) | (4.92,6.66) | (6.17,7.85) | (0.211,0.427) |
| Massachusetts | (2.19,2.4) | (20.1,28.4) | (24.3,44.7) | (0.018,0.026) | (1.24,3.71) | (4.69,6.2) | (3.44,4.11) | (0.193,0.339) |
| Michigan | (2.55,3.22) | (34.3,47.1) | (32,43.2) | (0.015,0.021) | (957.5,2872.49) | (3.21,4.05) | (5.69,6.78) | (0.121,0.2) |
| Minnesota | (1.71,2.04) | (20.4,51) | (12.2,138.6) | (0.008,0.011) | (118.62,355.86) | (2.87,3.86) | (7.09,9.05) | (0.241,0.434) |
| Mississippi | (2.04,3.02) | (106.3,260.8) | (4.9,10.5) | (0.011,0.015) | (165.05,495.16) | (2.33,3.03) | (4.5,5.44) | (0.231,0.307) |
| Missouri | (1.55,2.15) | (54.6,152.5) | (10.8,35.1) | (0.008,0.011) | (646.56,1939.67) | (1.54,2.01) | (6.97,8.51) | (0.097,0.173) |

(Continued.)

**Table 3.** (*Continued.*)

| State | $\mathcal{R}_0$ | $\psi$ | F. half-life (days) | IFR | $I_0$ | true case est. | T. case/r. case | D. cases 31 May (%) |
|---|---|---|---|---|---|---|---|---|
| Montana | (3.54,4.97) | (315.6,448.3) | (58,81.4) | (0.004,0.006) | (22.54,67.61) | (0.3,0.42) | (6.75,7.86) | (0.006,0.012) |
| Nebraska | (1.72,1.96) | (17.1,51.1) | (7.7,91.6) | (0.003,0.004) | (12.15,36.44) | (3.75,5.29) | (5.54,7.03) | (0.317,0.58) |
| Nevada | (1.85,2.32) | (38.3,59.6) | (25.4,53.5) | (0.005,0.007) | (300.34,901.03) | (2.33,3.03) | (8.59,10.7) | (0.088,0.158) |
| New Hampshire | (1.51,1.87) | (32.7,172.9) | (5.5,106.4) | (0.012,0.017) | (61.04,183.11) | (1.57,2.05) | (4.46,5.29) | (0.137,0.225) |
| New Jersey | (2.22,2.68) | (18.5,27.6) | (38.7,76.6) | (0.024,0.035) | (406.24,1218.72) | (4.29,5.93) | (2.47,3.01) | (0.097,0.201) |
| New Mexico | (1.53,1.99) | (56.5,168.1) | (11.3,94) | (0.016,0.022) | (134.42,403.27) | (1.05,1.35) | (2.89,3.42) | (0.075,0.131) |
| New York | (2.59,3.02) | (7,11.3) | (77.5,138.6) | (0.014,0.022) | (612.25,1834.75) | (8.57,12.78) | (4.9,6.18) | (0.06,0.168) |
| North Carolina | (2.02,2.67) | (96.6,258.8) | (4.9,12.9) | (0.004,0.006) | (195.51,586.53) | (2.2,2.8) | (8.69,10.89) | (0.206,0.287) |
| North Dakota | (1.4,1.87) | (50,179.1) | (15,138.6) | (0.011,0.015) | (47.41,142.24) | (0.77,1.02) | (2.47,2.83) | (0.058,0.097) |
| N. Mariana Is. | (2.16,6) | (302.6,719.3) | (80.1,138.6) | (0.010,0.012) | (62.8,150.0) | (0.29,0.48) | (10.78,14.46) | (0.001,0.004) |
| Ohio | (1.55,2.06) | (25.3,61.2) | (15.2,137) | (0.007,0.01) | (1700,5100) | (2.34,3.3) | (8.13,10.36) | (0.134,0.253) |
| Oklahoma | (1.43,2.11) | (51.6,106.7) | (25,60.4) | (0.005,0.008) | (1150.34,3451.01) | (1.24,1.74) | (8.31,10.36) | (0.044,0.08) |
| Oregon | (1.99,2.37) | (88.4,131.1) | (28.4,66.2) | (0.004,0.006) | (54.04,162.12) | (0.97,1.36) | (10.82,13.57) | (0.028,0.068) |
| Pennsylvania | (1.85,2.3) | (17.3,28.2) | (21.3,57.2) | (0.008,0.012) | (1250.05,3750) | (4.72,6.62) | (8.04,10.53) | (0.192,0.4) |
| Puerto Rico | (1.78,2.26) | (42.4,63.9) | (45.5,138.6) | (0.003,0.004) | (514.94,1403.78) | (1.58,2.06) | (16.12,19.23) | (0.032,0.061) |
| Rhode Island | (2.07,2.54) | (40.7,94.3) | (8.2,26.9) | (0.022,0.031) | (7.49,22.47) | (2.92,3.82) | (2.05,2.37) | (0.211,0.365) |
| South Carolina | (1.9,2.53) | (95.8,220.7) | (9.5,21.8) | (0.007,0.009) | (170.99,512.96) | (1.4,1.8) | (6.38,7.38) | (0.107,0.153) |
| South Dakota | (1.74,2.29) | (30.2,81.4) | (8.7,30.5) | (0.002,0.003) | (62.96,188.88) | (3.15,4.14) | (5.48,6.73) | (0.246,0.425) |
| Tennessee | (3.83,6) | (188,408.1) | (4.9,10.1) | (0.002,0.003) | (54,162) | (2.27,2.86) | (7.23,8.46) | (0.197,0.252) |
| Texas | (1.9,2.17) | (65.7,123.8) | (12.5,33.8) | (0.004,0.006) | (36.36,109.09) | (1.49,2.08) | (7.44,9.56) | (0.097,0.181) |
| Utah | (1.81,2.27) | (138,391.6) | (4.9,13.8) | (0.003,0.004) | (30.76,92.29) | (1.22,1.65) | (4.53,5.38) | (0.116,0.168) |
| Vermont | (1.94,2.77) | (67.7,90.8) | (85.3,138.6) | (0.006,0.009) | (186.09,532.3) | (1.07,1.5) | (7.59,9.17) | (0.006,0.013) |
| Virgin Islands | (1.47,2.17) | (84.7,131.7) | (65.4,138.6) | (0.006,0.009) | (37.02,102.17) | (0.68,1.02) | (10.48,13.48) | (0.009,0.018) |
| Virginia | (1.7,1.96) | (12.1,19.2) | (36.8,138.6) | (0.003,0.005) | (799.67,2399) | (4.54,6.13) | (9.85,12.49) | (0.227,0.413) |
| Washington | (2.06,2.26) | (52.2,76.9) | (37.4,82.6) | (0.009,0.013) | (1.45,4.34) | (1.59,2.11) | (5.96,7.5) | (0.028,0.069) |
| West Virginia | (3.25,4.19) | (146.1,214.9) | (24.9,34) | (0.004,0.006) | (12.17,36.52) | (0.87,1.19) | (8.5,10.54) | (0.054,0.076) |
| Wisconsin | (2.51,2.98) | (167.3,478.6) | (4.9,12.2) | (0.006,0.009) | (0.5,1.5) | (1.53,1.98) | (5.59,6.69) | (0.131,0.177) |
| Wyoming | (3.59,5.11) | (386.1,843) | (4.9,10.8) | (0.003,0.004) | (0.5,1.5) | (0.87,1.14) | (5.22,5.99) | (0.083,0.112) |

**Table 4.** Average relative error (ARE) for states and territories. All parameters are practically identifiable at a noise level of 70% except for North Dakota fatigue half-life.

| State | $\mathcal{R}_0$ | $\psi$ | F. half-life | IFR | $I_0$ | C. cases | T. case/ r. case | D. cases 31 May |
|---|---|---|---|---|---|---|---|---|
| Alabama | 12.33 | 29.45 | 26.32 | 7.30 | 30.53 | 6.10 | 2.25 | 8.67 |
| Alaska | 13.31 | 9.02 | 10.29 | 9.04 | 30.25 | 6.08 | 4.33 | 7.03 |
| Arizona | 2.97 | 24.80 | 22.57 | 7.07 | 30.32 | 5.40 | 3.39 | 7.52 |
| Arkansas | 9.33 | 12.82 | 7.41 | 11.55 | 30.03 | 5.80 | 3.37 | 6.08 |
| California | 2.97 | 22.49 | 23.99 | 6.51 | 34.21 | 5.67 | 5.45 | 7.54 |
| Colorado | 5.50 | 12.52 | 31.47 | 7.42 | 28.13 | 5.97 | 3.36 | 13.23 |
| Connecticut | 4.55 | 7.34 | 16.06 | 7.16 | 29.35 | 5.92 | 5.46 | 16.01 |
| Delaware | 5.05 | 11.49 | 49.22 | 8.23 | 27.99 | 6.91 | 5.05 | 14.79 |
| District of Columbia | 4.51 | 13.56 | 34.63 | 7.51 | 19.70 | 6.07 | 4.63 | 12.62 |
| Florida | 5.58 | 12.22 | 13.98 | 6.92 | 38.48 | 5.33 | 2.89 | 7.60 |
| Georgia | 5.04 | 12.68 | 13.44 | 8.67 | 39.81 | 5.03 | 3.44 | 6.24 |
| Guam | 2.37 | 8.35 | 8.40 | 8.30 | 33.01 | 6.76 | 6.48 | 24.31 |
| Hawaii | 5.61 | 9.74 | 7.98 | 11.81 | 32.42 | 10.74 | 6.99 | 24.55 |
| Idaho | 8.35 | 7.49 | 6.44 | 8.46 | 34.93 | 6.74 | 4.54 | 9.98 |
| Illinois | 2.46 | 13.15 | 34.10 | 9.19 | 33.39 | 7.08 | 5.67 | 13.28 |
| Indiana | 4.05 | 11.15 | 31.22 | 11.45 | 26.64 | 7.41 | 6.08 | 15.98 |
| Iowa | 4.18 | 17.41 | 67.89 | 7.18 | 25.81 | 6.36 | 3.14 | 11.69 |
| Kansas | 8.73 | 10.14 | 9.72 | 7.78 | 27.03 | 6.46 | 4.27 | 9.85 |
| Kentucky | 4.99 | 11.66 | 13.62 | 8.15 | 31.93 | 6.26 | 5.54 | 10.70 |
| Louisiana | 6.18 | 7.39 | 8.28 | 7.51 | 30.62 | 6.27 | 5.15 | 12.73 |
| Maine | 11.48 | 12.60 | 11.00 | 8.92 | 31.59 | 7.10 | 6.55 | 8.51 |
| Maryland | 3.74 | 11.19 | 25.68 | 10.11 | 27.72 | 6.62 | 6.02 | 13.46 |
| Massachusetts | 2.21 | 7.72 | 12.00 | 7.85 | 29.03 | 6.33 | 4.11 | 12.55 |
| Michigan | 5.47 | 6.41 | 5.92 | 7.28 | 34.98 | 5.53 | 4.09 | 9.85 |
| Minnesota | 3.71 | 19.56 | 40.71 | 8.57 | 26.65 | 6.75 | 5.17 | 10.92 |
| Mississippi | 8.32 | 19.04 | 14.10 | 6.97 | 36.30 | 5.62 | 3.52 | 6.43 |
| Missouri | 6.68 | 20.52 | 23.88 | 6.96 | 27.31 | 5.78 | 3.05 | 11.38 |
| Montana | 8.15 | 7.19 | 6.95 | 8.42 | 30.53 | 6.43 | 4.28 | 13.37 |
| Nebraska | 3.05 | 22.78 | 40.32 | 9.60 | 28.64 | 7.84 | 4.65 | 12.85 |
| Nevada | 5.03 | 9.67 | 14.59 | 8.21 | 28.09 | 5.79 | 5.43 | 12.09 |
| New Hampshire | 5.31 | 31.26 | 43.14 | 6.63 | 25.26 | 6.07 | 2.58 | 10.88 |
| New Jersey | 4.66 | 8.10 | 13.21 | 7.79 | 31.58 | 6.44 | 6.43 | 14.35 |
| New Mexico | 6.57 | 25.19 | 34.65 | 6.85 | 26.53 | 6.19 | 4.31 | 11.31 |
| New York | 4.21 | 9.76 | 17.78 | 8.95 | 29.05 | 9.23 | 7.20 | 23.04 |
| North Carolina | 6.00 | 20.32 | 20.45 | 6.89 | 25.39 | 5.69 | 4.47 | 7.01 |
| North Dakota | 6.53 | 30.22 | 129.83 | 6.73 | 26.78 | 5.90 | 3.31 | 12.23 |
| N. Mariana Islands | 13.15 | 29.75 | 23.74 | 29.01 | 25.15 | 18.89 | 10.62 | 20.27 |
| Ohio | 7.06 | 18.60 | 31.60 | 8.41 | 32.00 | 8.01 | 6.20 | 14.53 |
| Oklahoma | 9.52 | 13.97 | 14.97 | 9.63 | 25.74 | 7.90 | 6.27 | 13.66 |
| Oregon | 4.09 | 8.33 | 16.28 | 10.05 | 25.09 | 6.88 | 4.55 | 16.84 |

**Table 4.** (*Continued.*)

| State | $\mathcal{R}_0$ | $\psi$ | F. half-life | IFR | $I_0$ | C. cases | T. case/ r. case | D. cases 31 May |
|---|---|---|---|---|---|---|---|---|
| Pennsylvania | 4.78 | 10.11 | 18.57 | 9.20 | 26.65 | 7.96 | 6.18 | 15.07 |
| Puerto Rico | 5.98 | 9.64 | 53.89 | 11.43 | 22.45 | 7.01 | 4.96 | 21.18 |
| Rhode Island | 5.12 | 17.54 | 21.20 | 7.38 | 40.88 | 6.02 | 3.62 | 10.87 |
| South Carolina | 6.33 | 17.38 | 16.64 | 6.80 | 35.45 | 5.74 | 2.65 | 8.65 |
| South Dakota | 5.86 | 18.20 | 22.74 | 7.19 | 26.61 | 6.30 | 2.81 | 10.84 |
| Tennessee | 9.72 | 18.94 | 17.62 | 6.20 | 28.26 | 4.80 | 3.29 | 5.34 |
| Texas | 3.08 | 12.55 | 20.76 | 9.31 | 29.16 | 7.01 | 5.55 | 13.02 |
| Utah | 6.91 | 20.61 | 17.56 | 9.39 | 36.87 | 6.54 | 6.37 | 8.00 |
| Vermont | 7.60 | 6.35 | 15.07 | 7.35 | 20.21 | 6.54 | 3.56 | 24.70 |
| Virgin Islands | 9.91 | 10.65 | 26.21 | 10.60 | 22.79 | 8.63 | 5.64 | 15.26 |
| Virginia | 3.46 | 11.76 | 67.97 | 10.42 | 22.98 | 6.96 | 4.33 | 12.31 |
| Washington | 2.10 | 7.65 | 15.51 | 7.45 | 24.42 | 5.85 | 4.54 | 17.30 |
| West Virginia | 5.73 | 7.68 | 6.94 | 8.07 | 37.09 | 6.70 | 5.69 | 9.48 |
| Wisconsin | 3.81 | 24.00 | 23.55 | 7.02 | 29.44 | 6.39 | 4.49 | 5.89 |
| Wyoming | 7.56 | 18.17 | 20.33 | 6.79 | 33.95 | 5.47 | 2.45 | 6.55 |

**Table 5.** Approximate 95% confidence intervals, United States; quantity is said to be practically identifiable if its average relative error (ARE) is less than the noise level of 40%.

| quantity | 95% confidence interval | mean | median | ARE (%) |
|---|---|---|---|---|
| $\mathcal{R}_0$ | (2.21, 2.36) | 2.31 | 2.32 | 1.32 |
| $\psi$ | (44.7, 56.5) | 50.5 | 50.8 | 5.58 |
| fatigue half-life | (25.7, 29.9) | 27.7 | 27.6 | 3.25 |
| IFR | (0.0108, 0.0133) | 0.0121 | 0.0120 | 5.14 |
| $I_0$ | (17.6, 58.3) | 25.2 | 21.0 | 33.21 |
| cum. case (%) | (2.76, 3.37) | 3.03 | 3.03 | 4.92 |
| true cases/Rep. cases | (5.31, 6.30) | 5.78 | 5.81 | 2.33 |
| D. cases 31 May (%) | (0.14,0.18) | 0.16 | 0.16 | 5.25 |

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
