## [Peer Review File · Royal Society Open Science]

Review History

RSOS-210227.R0 (Original submission)

Review form: Reviewer 1

Is the manuscript scientifically sound in its present form?

No

Are the interpretations and conclusions justified by the results?

Yes

Is the language acceptable?

Yes

Do you have any ethical concerns with this paper?

No

Have you any concerns about statistical analyses in this paper?

No

Recommendation?

Reject

Comments to the Author(s)

General remarks to the authors:

The model and manuscript have promise but significant improvements will be needed to make strengthen the content.

I have included some comments below about the model, or at least the way it is written up in this submission. More broadly, however, I wonder what are the takeaways offered by this modeling effort that would not have been apparent to without the model? The paper would be greatly strengthened by presenting a clear scientific question that the article is designed to address. The current version reads like many results (some of which are almost certainly very interesting) without a clear purpose or conclusion that can be drawn from them.

Specific Comments:

About the model design and mathematical definitions:

The idea that the rate of entry to quarantine is proportional to the rate at which infections occur needs justification, especially if you are referring to ψ as a “lockdown factors”. Lockdowns are government policies that are mandated for entire populations simultaneously, and can should, be implemented even when the number of new infections remain low. The choice of a mass action term proportional to the mass action infection rate implies that the entry into lockdown would be gradual, and would be very slow if the slow if the state rate of infection is low (regardless, for example, of the rates in neighboring states).

The fact that the locked down population will vary at a rate always proportional to the infection rate is sure to introduce dynamic behavior artifacts into the simulations.

The saturating form of the ascertainment rate requires further justification. A connection between the given functional form and testing availability, testing result delays, and case rates is asserted, but not fully explained.

The form of the function $D(t)$ is almost certainly incorrect, as is the meaning ascribed to ξ . $D(t)$ is supposed to represent the number of deaths that occur during the outbreak, and ξ is defined as the infection fatality rate. If that is true, then as t goes to infinity, the integral in the definition must be equal to the cumulative number of infections, but that is not the case. t is used as a bound of integration and a variable in the integral. The term $t^{(\eta-1)}$ can be pulled out of the integral, and so long as the integrand is nonnegative (which it is), the limit of $D(t)$ as t goes to infinity will be infinity.

Also in the $D(t)$ function, μ is both a parameter (equal to 21 days) and the variable of integration. The effect is that a continuum of different gamma distributions are used over the interval of integration. I am not sure what this would mean.

I suspect that t in the gamma distribution and μ in $I(t-\mu)d\mu$ should both be replaced with a dummy variable s . Perhaps this was the intention.

In short, as presented, the parameter ξ is not the infection fatality rate. And the function $D(t)$ is not the time dependent number of deaths. Since $D(t)$ is used in the objective functional, this distorts the results and conclusions that can be drawn from the study.

In page 8 line 45, it is asserted that $C = \int_0^{\infty} I(t)/N dt$ is the final cumulative infection number. This is false. For example, if everyone were infected at $t=0$ and no one were infected or quarantined, the final cumulative infection number should be 1. But this integral would in fact be equal to T . The proper integral would be $\int_0^{\infty} \beta S(t)I(t) dt$. $1 - s_{\infty}$ should match this.

When public health strategies are discussed in the paper's conclusion, the factor of "low fatigue" or "high fatigue" is mentioned. How can the degree of fatigue be a strategic lever for public health officials to pull. This seems more like a characteristic of the population. Is this factor supposed to represent the rate at which lockdown restrictions are lifted by the government? In that case, a new name is needed for this parameter. Also, as with ψ , it is not clear why continuous rates of transition between the classes make sense. In short, what does "have low fatigue" mean from a public health policy perspective?

Other comments:

In Table 1, please provide units and either values or ranges of possible values.

On page 10, starting at line 47. References are made to "this first relationship" and "the last two relationships" but it is unclear exactly what is being referred to.

Furthermore, it is asserted that the law of large numbers explains something, but the subsequent explanation, while based on case counts being high, is not an example of the law of large numbers at all. The law states that in repeated experiments, the observed mean will approach the true mean. It is unclear how this applies here. Indeed, if the law of large numbers did apply, it would imply that the paper's results in states with lower case counts are not true observations, but rather are potentially simply the result of random fluctuations with small sample sizes.

Throughout the paper, there are small typographical and usage errors, such as using whom where who is appropriate, not capitalizing the word "States" in "United States".

There are also many instances where sweeping statements are made but the particulars are left vague. A clearer connection between the conclusions drawn and the evidence on which those conclusions are based would be helpful.

On page 15, a comparison is made between "is reduced by 90%" and "this difference is closer to a factor of five in baseline simulations". Such comparisons should be made in comparable terms. Are we talking about a 500% increase vs. a 90% decrease, or an 80% decrease vs. a 90% decrease?

Review form: Reviewer 2

Is the manuscript scientifically sound in its present form?

Yes

Are the interpretations and conclusions justified by the results?

Yes

Is the language acceptable?

Yes

Do you have any ethical concerns with this paper?

No

Have you any concerns about statistical analyses in this paper?

No

Recommendation?

Accept with minor revision (please list in comments)

Comments to the Author(s)

I enjoyed reading this well-written paper. This is an interesting mathematical modeling study with the goal of estimating the relationship between outbreak size and lockdown rate and reproduction number for the first wave of the COVID-19 pandemic in the US. The strengths of the paper lie in the connection of an innovative simple SEIR mathematical model with time-dependent ascertainment rate with COVID-19 epidemic curves of the USA at the state level. The patterns uncovered from the modeling study are interesting and seem robust. While I am not greatly concerned about parameter identifiability issues in this study since authors focused on estimating a few parameters and the model seems well constrained to the data, I would suggest to add measures of uncertainty/confidence intervals to their estimates (Table 2). Perhaps this could be done relatively easily with a parametric bootstrapping approach (see Chowell et al. <https://www.sciencedirect.com/science/article/pii/S2468042717300234>)

Decision letter (RSOS-210227.R0)

Dear Dr Gulbudak

The Editors assigned to your paper RSOS-210227 "Modeling COVID-19 outbreaks in United States with distinct testing, lockdown speed and fatigue rates" have now received comments from reviewers and would like you to revise the paper in accordance with the reviewer comments and any comments from the Editors. Please note this decision does not guarantee eventual acceptance.

Please submit your revised manuscript and required files (see below) no later than 21 days from today's (ie 31-Mar-2021) date. Note: the ScholarOne system will 'lock' if submission of the revision is attempted 21 or more days after the deadline. If you do not think you will be able to meet this deadline please contact the editorial office immediately.

on behalf of Dr Shigui Ruan (Associate Editor) and Glenn Webb (Subject Editor)
openscience@royalsociety.org

Associate Editor Comments to Author (Dr Shigui Ruan):

Please revise your manuscript carefully following all comments and suggestions, in particular those by the one who recommended rejection.

Reviewer comments to Author:
Reviewer: 1
Comments to the Author(s)

General remarks to the authors:

The model and manuscript have promise but significant improvements will be needed to make strengthen the content.

I have included some comments below about the model, or at least the way it is written up in this submission. More broadly, however, I wonder what are the takeaways offered by this modeling effort that would not have been apparent to without the model? The paper would be greatly strengthened by presenting a clear scientific question that the article is designed to address. The current version reads like many results (some of which are almost certainly very interesting) without a clear purpose or conclusion that can be drawn from them.

Specific Comments:

About the model design and mathematical definitions:

The idea that the rate of entry to quarantine is proportional to the rate at which infections occur needs justification, especially if you are referring to ψ as a “lockdown factors”. Lockdowns are government policies that are mandated for entire populations simultaneously, and can should, be implemented even when the number of new infections remain low. The choice of a mass action term proportional to the mass action infection rate implies that the entry into lockdown would be gradual, and would be very slow if the slow if the state rate of infection is low (regardless, for example, of the rates in neighboring states).

The fact that the locked down population will vary at a rate always proportional to the infection rate is sure to introduce dynamic behavior artifacts into the simulations.

The saturating form of the ascertainment rate requires further justification. A connection between the given functional form and testing availability, testing result delays, and case rates is asserted, but not fully explained.

The form of the function $D(t)$ is almost certainly incorrect, as is the meaning ascribed to ξ . $D(t)$ is supposed to represent the number of deaths that occur during the outbreak, and ξ is defined as the infection fatality rate. If that is true, then as t goes to infinity, the integral in the definition must be equal to the cumulative number of infections, but that is not the case. t is used as a bound of integration and a variable in the integral. The term $t^{\eta-1}$ can be pulled out of the integral, and so long as the integrand is nonnegative (which it is), the limit of $D(t)$ as t goes to infinity will be infinity.

Also in the $D(t)$ function, μ is both a parameter (equal to 21 days) and the variable of integration. The effect is that a continuum of different gamma distributions are used over the interval of integration. I am not sure what this would mean.

I suspect that t in the gamma distribution and μ in $I(t-\mu)d\mu$ should both be replaced with a dummy variable s . Perhaps this was the intention.

In short, as presented, the parameter ξ is not the infection fatality rate. And the function $D(t)$ is not the time dependent number of deaths. Since $D(t)$ is used in the objective functional, this distorts the results and conclusions that can be drawn from the study.

In page 8 line 45, it is asserted that $C = \int_0^{\infty} I(t)/N dt$ is the final cumulative infection number. This is false. For example, if everyone were infected at $t=0$ and no one were infected or quarantined, the final cumulative infection number should be 1. But this integral would in fact be equal to T . The proper integral would be $\int_0^{\infty} \beta S(t)I(t) dt$. $1-s_{\infty}$ should match this.

When public health strategies are discussed in the paper's conclusion, the factor of "low fatigue" or "high fatigue" is mentioned. How can the degree of fatigue be a strategic lever for public health officials to pull. This seems more like a characteristic of the population. Is this factor supposed to represent the rate at which lockdown restrictions are lifted by the government? In that case, a new name is needed for this parameter. Also, as with ψ , it is not clear why continuous rates of transition between the classes make sense. In short, what does "have low fatigue" mean from a public health policy perspective?

Other comments:

In Table 1, please provide units and either values or ranges of possible values.

On page 10, starting at line 47. References are made to "this first relationship" and "the last two relationships" but it is unclear exactly what is being referred to.

Furthermore, it is asserted that the law of large numbers explains something, but the subsequent explanation, while based on case counts being high, is not an example of the law of large numbers at all. The law states that in repeated experiments, the observed mean will approach the true mean. It is unclear how this applies here. Indeed, if the law of large numbers did apply, it would imply that the paper's results in states with lower case counts are not true observations, but rather are potentially simply the result of random fluctuations with small sample sizes.

Throughout the paper, there are small typographical and usage errors, such as using whom where who is appropriate, not capitalizing the word “States” in “United States”.

There are also many instances where sweeping statements are made but the particulars are left vague. A clearer connection between the conclusions drawn and the evidence on which those conclusions are based would be helpful.

On page 15, a comparison is made between “is reduced by 90%” and “this difference is closer to a factor of five in baseline simulations”. Such comparisons should be made in comparable terms. Are we talking about a 500% increase vs. a 90% decrease, or an 80% decrease vs. a 90% decrease?

Reviewer: 2

Comments to the Author(s)

I enjoyed reading this well-written paper. This is an interesting mathematical modeling study with the goal of estimating the relationship between outbreak size and lockdown rate and reproduction number for the first wave of the COVID-19 pandemic in the US. The strengths of the paper lie in the connection of an innovative simple SEIR mathematical model with time-dependent ascertainment rate with COVID-19 epidemic curves of the USA at the state level. The patterns uncovered from the modeling study are interesting and seem robust. While I am not greatly concerned about parameter identifiability issues in this study since authors focused on estimating a few parameters and the model seems well constrained to the data, I would suggest to add measures of uncertainty/confidence intervals to their estimates (Table 2). Perhaps this could be done relatively easily with a parametric bootstrapping approach (see Chowell et al. <https://www.sciencedirect.com/science/article/pii/S2468042717300234>)

===PREPARING YOUR MANUSCRIPT===

If you have been asked to revise the written English in your submission as a condition of publication, you must do so, and you are expected to provide evidence that you have received language editing support. The journal would prefer that you use a professional language editing service and provide a certificate of editing, but a signed letter from a colleague who is a native

speaker of English is acceptable. Note the journal has arranged a number of discounts for authors using professional language editing services (<https://royalsociety.org/journals/authors/benefits/language-editing/>).

===PREPARING YOUR REVISION IN SCHOLARONE===

<https://royalsociety.org/journals/authors/author-guidelines/#supplementary-material> to include a suitable title and informative caption. An example of appropriate titling and captioning may be found at https://figshare.com/articles/Table_S2_from_Is_there_a_trade-

off_between_peak_performance_and_performance_breadth_across_temperatures_for_aerobic_sc
ope_in_teleost_fishes_/3843624.

Author's Response to Decision Letter for (RSOS-210227.R0)

See Appendix A.

RSOS-210227.R1 (Revision)

Review form: Reviewer 1

Is the manuscript scientifically sound in its present form?

Yes

Are the interpretations and conclusions justified by the results?

Yes

Is the language acceptable?

Yes

Do you have any ethical concerns with this paper?

No

Have you any concerns about statistical analyses in this paper?

No

Recommendation?

Accept as is

Comments to the Author(s)

I believe that you have addressed the major concerns raised in the first round of reviews, and I thank you for taking those comments seriously and doing the work to make these substantial changes.

Decision letter (RSOS-210227.R1)

Dear Dr Gulbudak,

It is a pleasure to accept your manuscript entitled "Modeling COVID-19 outbreaks in United States with distinct testing, lockdown speed and fatigue rates" in its current form for publication in Royal Society Open Science. The comments of the reviewer(s) who reviewed your manuscript are included at the foot of this letter.

COVID-19 rapid publication process:

We are taking steps to expedite the publication of research relevant to the pandemic. If you wish, you can opt to have your paper published as soon as it is ready, rather than waiting for it to be published the scheduled Wednesday.

This means your paper will not be included in the weekly media round-up which the Society sends to journalists ahead of publication. However, it will still appear in the COVID-19 Publishing Collection which journalists will be directed to each week (<https://royalsocietypublishing.org/topic/special-collections/novel-coronavirus-outbreak>).

If you wish to have your paper considered for immediate publication, or to discuss further, please notify openscience_proofs@royalsociety.org and press@royalsociety.org when you respond to this email.

on behalf of Dr Shigui Ruan (Associate Editor) and Glenn Webb (Subject Editor)
openscience@royalsociety.org

Reviewer comments to Author:

Reviewer: 1

Comments to the Author(s)

I believe that you have addressed the major concerns raised in the first round of reviews, and I thank you for taking those comments seriously and doing the work to make these substantial changes.

Response to Editor and Reviewer Comments

Dear Editor and Reviewers,

Thank you for the reviews of our manuscript, which led to substantial revisions and improvements in the work. In particular, in light of comments by reviewer 1, we highlight that we added extensive justification of the model term $\psi\lambda(t)$, representing infection-level dependent self-quarantine/lockdown rate, including several references and a new figure 7 (pg. 13) showing the correspondence between model compartment S_q , (self) quarantined individuals, and decrease in mobility using data from [1] for four example states (Connecticut, New Jersey, Louisiana and Texas) during the time period for which the mobility data is available (up to April 20, 2020). We also have reformulated the death compartment and infection fatality ratio (IFR), and subsequently re-calibrated all model fits and updated all simulation figures, leading to correct interpretation of corresponding variables/parameters and improved model fits. We also carry out further uncertainty quantification we have generated confidence intervals for the key model parameters across all 55 considered states and territories, using an approach similar to reference [3] provided by reviewer 2.

Below is a detailed point-by-point response (in blue) to each reviewer comment with explanation of all significant edits in the revised manuscript

reviewer one

- (i) The idea that the rate of entry to quarantine is proportional to the rate at which infections occur needs justification, especially if you are referring to ψ as a “lockdown factors”. Lockdowns are government policies that are mandated for entire populations simultaneously, and can should, be implemented even when the number of new infections remain low. The choice of a mass action term proportional to the mass action infection rate implies that the entry into lockdown would be gradual, and would be very slow if the slow if the state rate of infection is low (regardless, for example, of the rates in neighboring states).

We add a paragraph under model (2.1) to justify this formulation. To explain our justification, note that the product of ψ with force of infection is a phenomenological relation between lockdown/self-quarantine rates and current infection levels. Indeed, several works [1,4,6,8,11] have shown that population activity measures (e.g. mobility, economic transactions, percentage of people staying/working at home) were primarily driven by individual reaction to media and perceived risk tied to COVID-19 case incidence, and secondarily influenced by government mandates. Indeed our model captures the social nature of ψ in the positive relationship between ψ and \mathcal{R}_t . \mathcal{R}_0 is an indicator of stronger force of infection and this relationship is likely explained by public reaction to resulting perceived risk. State lockdown orders are also inherently related to case counts, but may be enacted in a temporally discrete manner with other factors affecting their proclamation. Although reporting accuracy and delays in response complicate the relationship of human behavioral changes and government action with raw infection incidence, our formulation offers a simple measure of population self-quarantine rate relative to case incidence (see [2] for re-scaled rates accounting for reporting delay). Other approaches for capturing large-scale social distancing/self-quarantine in populations have been utilized, such as assuming time-dependent transmission/contact rates ($\beta(t)$) [5,10] or considering constant rate of susceptible transition to quarantined state [7]. While there are advantages/disadvantages to each modeling approach, we contend that our nonlinear social distancing rate captures a contagion-like behavioral response to rising infected cases, and allows us to derive novel formulae for final and peak outbreak size (see results section of manuscript). Furthermore by tying quarantine to new infection rate, we capture the observed rapid

large scale response of varying strength across states, which saturates and wanes as cases drop and fatigue sets in, mimicking mobility data from [1] (see Figure 7). We agree that the notation “lockdown factor” may be confusing, especially since in original manuscript we highlighted time of government stay-at-home orders in some figures. So, we clearly explained now that by lockdown we mean an observed population-wide social behavior change caused by individual and government reaction to incidence, and removed highlighted government stay-at-home orders from all figures.

- (ii) The fact that the locked down population will vary at a rate always proportional to the infection rate is sure to introduce dynamic behavior artifacts into the simulations. While it is true that entry into quarantine is tied to force of infection, exit rate from quarantine, (model parameter α) is not. Therefore the population in quarantine is in fact not always proportional to force of infection, rather only the rate of entry. We have edited the manuscript to indicate this. In addition as mentioned in response to the previous comment, our locked down population component trajectories are consistent with mobility data, which is highlighted in for four example states in the newly inserted Fig. 7 (on pg. 13 in main text).
- (iii) The saturating form of the ascertainment rate requires further justification. A connection between the given functional form and testing availability, testing result delays, and case rates is asserted, but not fully explained. We note that our model captures test positivity well, and that our ascertainment rate is directly responsible for this, as the two have an inversely proportional relationship. We have modified Figs. 4,7 of the main text and SI Figs. S1-S14 by replacing raw testing data with ascertainment rate on the second axis to visually highlight this relationship. For Connecticut we observe that resulting cumulative ascertainment ratio,

$$A_t = \frac{\int_0^t (\rho(t)/T) I(t) dt}{\int_0^t \lambda(t) S(t) dt},$$

matches very well with that provided in [9] (new SI Fig. S15). Finally, while this form of $\rho(t)$ is a phenomenological relationship with $\tau(t)$ and there is no formal derivation, we can sketch further justification of a saturating $\rho(t)$ when the number of tests, $\tau(t)$, increases with t , as observed over the time frame. Indeed, we propose the informal differential equation for $\rho(t)$:

$$\frac{d\rho}{dt} = a\tau(t) - b\rho(t) - \frac{\rho(t)}{T} \frac{d}{dt} \left(\frac{I(t)}{\tau(t)} \right)$$

reflecting increase due to testing availability (at first increasing with testing but linear decreasing as higher proportion tested results in complacency and/or longer waits) and the change in $\rho(t)$ from the flux in positivity rate, respectively. Under appropriate conditions, then $\rho(t)$ will increase to a limit in saturating form as $t \rightarrow \infty$.

- (iv) The form of the function $D(t)$ is almost certainly incorrect, as is the meaning ascribed to ξ . $D(t)$ is supposed to represent the number of deaths that occur during the outbreak, and ξ is defined as the infection fatality rate. If that is true, then as t goes to infinity, the integral in the definition must be equal to the cumulative number of infections, but that is not the case. t is used as a bound of integration and a variable in the integral. The term $t^{(\eta-1)}$ can be pulled out of the integral, and so long as the integrand is nonnegative (which it is), the limit of $D(t)$ as t goes to infinity will be infinity.

Also in the $D(t)$ function, μ is both a parameter (equal to 21 days) and the variable of integration. The effect is that a continuum of different gamma distributions are used over the interval of integration. I am not sure what this would mean.

I suspect that t in the gamma distribution and μ in $I(t - \mu)d\mu$ should both be replaced with a dummy variable s . Perhaps this was the intention.

In short, as presented, the parameter ξ is not the infection fatality rate. And the function $D(t)$ is not the time dependent number of deaths. Since $D(t)$ is used in the objective functional, this distorts the results and conclusions that can be drawn from the study.

We agree that the equation for deaths is incorrect as stated in the original version of the manuscript, and did not correctly reflect how this value was calculated in the program files for the original draft of the manuscript:

```

for j = 2:length(tD)
    mdd(j-1)=(deathdist((2:j),ag,bg)...
    -deathdist((1:j-1),ag,bg))*(infs(j-1:-1:1));
    z2(j) =z2(j-1)+mdd(j-1);
end
z2 = xi*z2;

```

the form of D in the original manuscript should have thus been written as:

$$D(t) = \xi \int_0^t \frac{a^{\eta-1} \exp(-\eta a/\mu)}{\Gamma(\eta)(\mu/\eta)^\eta} I(t-a) da$$

. The reviewer is also correct in that, as originally stated ξ did not represent IFR. With this in mind in our revision with have formulated

$$D(t) = \xi \int_0^t \frac{a^{\eta-1} \exp(-\eta a/\mu)}{\Gamma(\eta)(\mu/\eta)^\eta} I(t-a) S(t-a) da,$$

after which ξ does now represent IFR. We have updated model diagram in Fig. 1 to show the cumulative amount of dead individuals, $D(t)$, given by the proportion ξ of infection incidence and gamma distributed time (after infection) until death with fixed mean of $\mu = 21$ days (based on ref. [8] in main text). Indeed with this new formulation, we retain the the cumulative number of infections (multiplied by ξ) as $t \rightarrow \infty$ in the integral. Furthermore, we rerun the fitting algorithm for all states to calibrate the new calculation of $D(t)$ to the cumulative death data. While the general relationships between parameters hold, with only slight variation in correlation, and most parameter values are similar to the original fit values (all tables, simulation studies, fitting plots, etc. have been updated to reflect new fits) this formulation for $D(t)$ results in, roughly speaking, a 90% reduction in fit ψ value (in the scenario with no lockdown fatigue ($\alpha = 0$) this corresponds to a 10 day difference in time until 50% of the population enters self-quarantine). We feel these new ψ values are more realistic, as evidenced by improved model fit to data, and we thank the reviewer for the comment which led to this modification.

- (v) In page 8 line 45, it is asserted that $C = \int_0^\infty I(t)/N dt$ is the final cumulative infection number. This is false. For example, if everyone were infected at $t=0$ and no one were infected or quarantined, the final cumulative infection number should be 1. But this integral would in fact be equal to T . The proper integral would be $\int_0^\infty \beta S(t) I(t) dt$. $1 - s_\infty$ should match this.

The reviewer is correct and the equation has been updated accordingly

- (vi) When public health strategies are discussed in the paper's conclusion, the factor of "low fatigue" or "high fatigue" is mentioned. How can the degree of fatigue be a strategic lever for public health officials to pull. This seems more like a characteristic of the population. Is this factor supposed to represent the rate at which lockdown restrictions are lifted by the government? In that case, a new name is needed for this parameter. Also, as with ψ , it is not clear why continuous rates of transition between the classes make sense. In short, what does "have low fatigue" mean from a public health policy perspective?
 α is simply the exit rate from self-quarantine measuring the population level lockdown fatigue, in other words the tendency for individuals and and government to revert to normal regardless of infection level after a certain amount of time. That being said because of the public reaction to infection by self-quarantine, the infection levels will inevitably drop (dependent on magnitude of ψ), so population will revert to normalcy when the disease is more under control than before lockdown/self-quarantine (given the range of ψ and α parameters estimated in our fitting work). As such while public officials

cannot directly control α there are possible government actions which could influence it. Indeed how the top levels of government react to an epidemic are crucial to success of the response, with one study finding early information focused actions and emergency declarations had significant impact on social behaviour [4] We have added a description in the model section and paragraph to the discussion section of manuscript to emphasize these points about α , and suggest that testing continue to detect subsequent waves and once such a wave is detected information be rapidly disseminated emphasizing the importance of both quick response and duration of self-quarantine. We also feel that $\log(2)/\alpha$ (fatigue half-life), the amount of time it takes for 50% of the population which entered self-quarantine to return to normalcy (in days) provides a more readily interpreted quantity in terms of public health implications. As such we have also changed all references of "low fatigue" to "fatigue half-life of at least 60 days" and "high fatigue" to "fatigue half-life of less than 15 days", and "intermediate fatigue" to "fatigue half-life between 15 and 60 days"

- (vii) In Table 1, please provide units and either values or ranges of possible values. The table has been appropriately updated.
- (viii) On page 10, starting at line 47. References are made to "this first relationship" and "the last two relationships" but it is unclear exactly what is being referred to. This section has been rewritten to clarify what is meant.
- (ix) Furthermore, it is asserted that the law of large numbers explains something, but the subsequent explanation, while based on case counts being high, is not an example of the law of large numbers at all. The law states that in repeated experiments, the observed mean will approach the true mean. It is unclear how this applies here. Indeed, if the law of large numbers did apply, it would imply that the paper's results in states with lower case counts are not true observations, but rather are potentially simple the result of random fluctuations with small sample sizes. The reviewer is correct, and this paragraph has been revised accordingly.
- (x) Throughout the paper, there are small typographical and usage errors, such as using whom where who is appropriate, not capitalizing the word "States" in "United States". We have thoroughly reviewed the paper for typographical and grammatical errors. We appreciate the reviewer taking time to point this out.
- (xi) There are also many instances where sweeping statements are made but the particulars are left vague. A clearer connection between the conclusions drawn and the evidence on which those conclusions are based would be helpful. We have gone through the discussion portion of the manuscript and added references to figures and tables upon which our conclusions are based. Additionally we have significantly revised figure 9, and added new figure 10 (containing information previously in the SI) in the main text of the manuscript, as these visually represent and support many of the conclusions in the discussion section of the paper.
- (xii) On page 15, a comparison is made between "is reduced by 90%" and "this difference is closer to a factor of five in baseline simulations". Such comparisons should be made in comparable terms. Are we talking about a 500% increase vs. a 90% decrease, or an 80% decrease vs. a 90% decrease? this section has been rewritten to clarify what it was intended to convey with all quantities clearly described in terms of relative change (as percent).

reviewer two

- (i) I enjoyed reading this well-written paper. This is an interesting mathematical modeling study with the goal of estimating the relationship between outbreak size and lockdown rate and reproduction number for the first wave of the COVID-19 pandemic in the US. The strengths of the paper lie in the connection of an innovative simple SEIR mathematical model with time-dependent ascertainment rate with COVID-19 epidemic

curves of the USA at the state level. The patterns uncovered from the modeling study are interesting and seem robust. While I am not greatly concerned about parameter identifiability issues in this study since authors focused on estimating a few parameters and the model seems well constrained to the data, I would suggest to add measures of uncertainty/confidence intervals to their estimates (Table 2). Perhaps this could be done relatively easily with a parametric bootstrapping approach (see Chowell et al. <https://www.sciencedirect.com/science/article/pii/S2468042717300234>) **We thank the reviewer for this suggestion. Tables with confidence intervals as ARE (average relative error) similar to the method in the provided source, though assuming normal error instead of Poisson because the former generates daily case (death) data much less variable the the underlying original data-sets (see SI Fig. 17)**

We have also included tracked-changes versions of both our manuscript and the supplementary information, which contains colored text and marks indicating our edits.

Sincerely yours,

J.C. Macdonald, C. Browne, and H. Gulbudak

References

1. H. S. BADR, H. DU, M. MARSHALL, E. DONG, M. M. SQUIRE, AND L. M. GARDNER, *Association between mobility patterns and covid-19 transmission in the usa: a mathematical modelling study*, *The Lancet Infectious Diseases*, 20 (2020), pp. 1247–1254, doi:[https://doi.org/10.1016/S1473-3099\(20\)30553-3](https://doi.org/10.1016/S1473-3099(20)30553-3), <https://www.sciencedirect.com/science/article/pii/S1473309920305533>.
2. C. BROWNE, H. GULBUDAK, AND J. MACDONALD, *Differential impacts of contact tracing and lockdowns on outbreak size in covid-19 model applied to china*, *Journal of Theoretical Biology* (submitted), (2020), <https://www.medrxiv.org/content/medrxiv/early/2020/06/15/2020.06.10.20127860.full.pdf>.
3. G. CHOWELL, *Fitting dynamic models to epidemic outbreaks with quantified uncertainty: A primer for parameter uncertainty, identifiability, and forecasts*, *Infectious Disease Modeling*, (2017).
4. S. GUPTA, T. D. NGUYEN, F. L. ROJAS, S. RAMAN, B. LEE, A. BENTO, K. I. SIMON, AND C. WING, *Tracking public and private responses to the covid-19 epidemic: Evidence from state and local government actions*, Working Paper 27027, National Bureau of Economic Research, April 2020, doi:[10.3386/w27027](https://doi.org/10.3386/w27027), <http://www.nber.org/papers/w27027>.
5. S. LAI, N. W. RUKTANONCHAI, L. ZHOU, O. PROSPER, W. LUO, J. R. FLOYD, A. WESOLOWSKI, M. SANTILLANA, C. ZHANG, X. DU, ET AL., *Effect of non-pharmaceutical interventions to contain covid-19 in china*, *Nature*, (2020), pp. 1–7.
6. M. LEE, J. ZHAO, Q. SUN, Y. PAN, W. ZHOU, C. XIONG, AND L. ZHANG, *Human mobility trends during the early stage of the covid-19 pandemic in the united states*, *PLOS ONE*, 15 (2020), pp. 1–15, doi:[10.1371/journal.pone.0241468](https://doi.org/10.1371/journal.pone.0241468), <https://doi.org/10.1371/journal.pone.0241468>.
7. B. F. MAIER AND D. BROCKMANN, *Effective containment explains subexponential growth in recent confirmed covid-19 cases in china*, *Science*, 368 (2020), pp. 742–746.
8. A. SHERIDAN, A. L. ANDERSEN, E. T. HANSEN, AND N. JOHANNESSEN, *Social distancing laws cause only small losses of economic activity during the covid-19 pandemic in scandinavia*, *Proceedings of the National Academy of Sciences*, 117 (2020), pp. 20468–20473, doi:[10.1073/pnas.2010068117](https://doi.org/10.1073/pnas.2010068117), <https://www.pnas.org/content/117/34/20468>, arXiv:<https://arxiv.org/abs/2006.07211>.
9. K. SHIODA AND ET AL., *Estimating the cumulative incidence of sars-cov-2 infection % and the infection fatality ratio in light of waning antibodies*, pre-print, (2020), <https://www.medrxiv.org/content/10.1101/2020.11.13.20231266v1.full.pdf>.
10. B. TANG, F. XIA, S. TANG, N. L. BRAGAZZI, Q. LI, X. SUN, J. LIANG, Y. XIAO, AND J. WU, *The evolution of quarantined and suspected cases determines the final trend of the 2019-ncov epidemics based on multi-source data analyses*, Available at SSRN 3537099, (2020).
11. C. XIONG, S. HU, M. YANG, H. YOUNES, W. LUO, S. GHADER, AND L. ZHANG, *Mobile device location data reveal human mobility response to state-level stay-at-home orders during the covid-19 pandemic in the usa*, *Journal of The Royal Society Interface*, 17 (2020), p. 20200344, doi:[10.1098/rsif.2020.0344](https://doi.org/10.1098/rsif.2020.0344), <https://royalsocietypublishing.org/doi/abs/10.1098/rsif.2020.0344>, arXiv:<https://arxiv.org/abs/2006.07211>.